# MagicDec: Breaking the Latency-Throughput Tradeoff for Long Context Generation with Speculative Decoding

**Ranajoy Sadhukhan**[1*]   **Jian Chen**[1*]   **Zhuoming Chen**[1]   **Vashisth Tiwari**[1]   **Ruihang Lai**[1]
**Jinyuan Shi**[2]   **Ian En-Hsu Yen**[2]   **Avner May**[3]   **Tianqi Chen**[1]   **Beidi Chen**[1]
[1]Carnegie Mellon University   [2]Moffett AI   [3]Together AI

## Abstract

Large Language Models (LLMs) have become more prevalent in long-context applications such as interactive chatbots, document analysis, and agent workflows, but it is challenging to serve long-context requests with low latency and high throughput. Speculative decoding (SD) is a widely used technique to reduce latency losslessly, but the conventional wisdom suggests that its efficacy is limited to small batch sizes. In **MagicDec**, we show that surprisingly SD can achieve speedup even for a high throughput inference regime for *moderate to long sequences*. More interestingly, an intelligent drafting strategy can achieve *better speedup with increasing batch size* based on our rigorous analysis. MagicDec first identifies the bottleneck shifts with increasing batch size and sequence length, and uses these insights to deploy SD more effectively for high throughput inference. We leverage draft model with sparse KV cache to address the KV bottleneck, which scales with both sequence length and batch size. Additionally, we propose a theoretical model to select the optimal drafting strategy for maximum speedup. Our work highlights the broad applicability of speculative decoding in long-context serving, as it can *enhance throughput and reduce latency without compromising accuracy*. For moderate to long sequences, we demonstrate up to **2.51x** speedup for `LLaMA-3.1-8B` when serving batch sizes ranging from 32 to 256 on various types of hardware and tasks.

## 1 Introduction

The emergence of extremely long-context Large Language Models (LLMs) (AI@Meta, 2024; QwenTeam, 2024; Liu et al., 2023) has led to the popularity of long-context applications such as retrieval augmented generation (Lewis et al., 2021), code generation (AWS, 2024; Chen et al., 2021) and document summarization. Low latency and high throughput are both crucial for serving these long-context LLMs – low latency ensures a positive user experience in interactive applications like chatbots (Achiam et al., 2023; Deepmind, 2024), while high throughput amortizes serving costs.

However, optimizing both latency and throughput in LLM serving presents significant challenges. Speculative decoding (SD)(Leviathan et al., 2022; Xia et al., 2023; Chen et al., 2023) can reduce latency by using a smaller model to predict multiple tokens ahead followed by verification by the target model. But this approach becomes inefficient with large batch sizes because of increased verification cost(Liu et al., 2024a; Su et al., 2023), as shown in Fig. 7a. For small batches, the main performance bottleneck is the parameter loading cost, which can be amortized by the verification process across the tokens to be verified at the expense of increased computation. However, with large batches, LLMs become compute bound, making verification significantly costly because of its compute-hungry nature. Additionally, if the smaller model's predictions do not align well with the target model, frequent costly verifications are needed. Consequently, the usage of SD in high batch size regime is discouraged by existing works(Liu et al., 2024a; Su et al., 2023; Miao et al., 2023). On the other hand, techniques like (Kwon et al., 2023; Yu et al., 2022; Agrawal et al., 2024b) improve throughput by accommodating larger batches, but at the cost of increased token-wise latency. While techniques such as quantization, pruning and KV cache eviction (Frantar et al., 2023; Xiao et al., 2024a; Hooper et al., 2024; Ma et al., 2023; Sun et al., 2024b) can improve both throughput and latency, they typically result in lower quality model outputs.

Based on these challenges, we pose the following question:

---

*Equal contribution

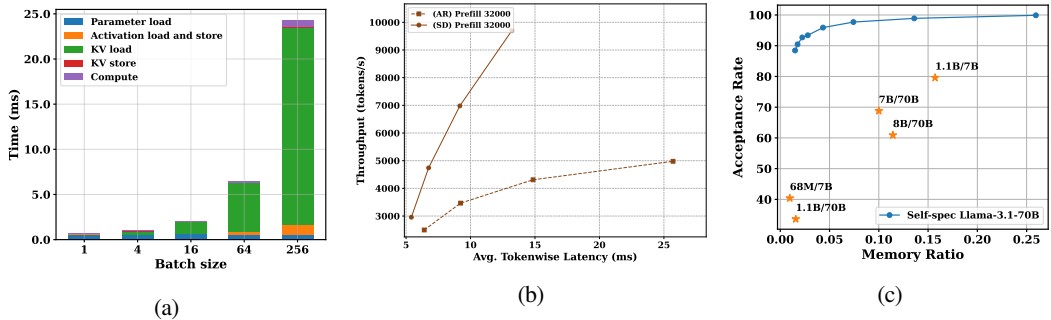

(a)         (b)         (c)

Figure 1: (a) Time breakdown of `LLaMA-3.1-8B` vs batch size (input length=16384, hardware=8xH100s). (b) Throughput of autoregressive decoding and StreamingLLM-based self-speculation of `LLaMA-3.1-8B` against per-token latency for prompt length of 32k. (c) Draft token acceptance rate comparison for `LLaMA-3.1-70B`. Self-speculation using Top-k attention achieves a much higher acceptance rate than other draft-target pairs, even with lower memory ratio. The x-axis represents the ratio of draft memory footprint to target memory footprint.

> *Can we simultaneously improve **throughput** and **latency** without sacrificing **accuracy**, particularly for long sequences?*

We answer with a resounding yes! For large batches of long sequences, we show that SD can be used more effectively to improve both throughput and latency without degradation of the output quality. We base our hypothesis on the following interesting insights:

(1) *KV Cache Is The Dominant Bottleneck In Large batch size Long-context Regime*: In long-context and large batch size regime, the KV cache outgrows the memory footprint of the model parameter and continues to increase with batch size. Computation also increases with batch size, but due to the high peak FLOPS-to-memory bandwidth ratio of modern GPUs, the KV loading time increases much more than former for larger batches, making LLM inference more memory-bound, as shown in Fig. 1a (Yuan et al., 2024).

(2) *SD Can Improve Throughput Only Beyond a Critical Sequence Length*: While existing research(Liu et al., 2024a; Su et al., 2023) suggests that SD is inefficient for large batches due to high verification costs, this limitation only applies to very short sequences. Because with short sequences, increasing the batch size makes computational costs the primary bottleneck, which is prohibitive for an efficient verification process. However, once sequences exceed a certain critical length (which varies based on the model and hardware), the KV loading cost becomes the dominant factor, even for large batches. At this point, SD becomes effective again because the computational overhead of verification becomes less significant compared to the KV loading costs, which can be amortized across the tokens to be verified.

(3) *Compressed KV Cache Enables More Efficient Speculation*: Token acceptance rate is crucial for SD in large batch processing to minimize costly verification steps. Our research found that compressing the Draft KV cache leads to higher acceptance rates than compressing model weights. To evaluate model compression only, we test different draft-target pairs on PG-19 (Rae et al., 2019) sequences of length only 256, to restrict the KV cache impact. For KV compression, we tested `LLaMA-3.1-70B` on longer sequences (4,000-100,000 tokens)[1] using Top-K selection for KV sparsification. Fig. 1c illustrates that model compression alone is unable to reach 90% acceptance rate, while KV compression achieved significantly higher rates under similar memory constraints. This advantage becomes even more significant with larger batch sizes, offering a promising new direction for improving the batch-processing efficiency speculative decoding.

Building upon these insights, our work **MagicDec** illustrates that SD can improve speedup even for large batches by utilizing KV compression, contrary to prior belief. As shown in Fig. 1b, under long context-length, compressed KV-based self-speculation can improve throughput and latency at the same time in all spectrum, without hurting generation quality. Furthermore, MagicDec evaluates different KV compression-based drafting methods to determine the optimal approach based on the specific model, hardware, and task requirements. We structure the paper as follows.

- In Section 3.1, we theoretically analyze the factors that decide the efficiency of speculative decoding. Section 3.2 discusses how the performance bottlenecks in LLM inference shift with batch size and sequence length, and what are its implications on SD's batch-processing efficiency. In the light of this study, we discuss the challenges involved with conventional SD in large batch setting and how it can be overcome

---

[1] batch size is set to 1 to nullify the effect of batch size on KV cache size

by KV sparsification based drafting. Additionally, we introduce the concept of the critical sequence length beyond which SD can achieve higher speedups for larger batches contrary to prior studies (Liu et al., 2024a; Su et al., 2023; Miao et al., 2023).

- In Section 3.3, we show why compressing the KV cache is crucial for effective batch processing. Our experimental results demonstrate that this approach achieves higher acceptance rates and, consequently, better batch performance compared to using parameter-efficient draft models. Section 4.4 discusses the trade-off between draft cost and acceptance rate for different static and dynamic KV sparsification algorithms on different kinds of tasks.

- Finally, in Section 5 we provide a comprehensive empirical evaluation across different hardware setups and tasks to show the effectiveness of our theoretical analysis and method. We demonstrate that our approach achieves a **2.51x speedup** in large batch settings for `LLaMA-3.1-8B` on 8xH100 GPUs, significantly improving both throughput and latency over traditional autoregressive decoding ( §5).

## 2 RELATED WORKS

Numerous efforts have been made to improve the latency and throughput of LLMs. Methods like Flash decoding (Dao, 2023), and Flash decoding++(Hong et al., 2023) have performed system optimizations to improve latency. KV compression methods (Li et al., 2024; Gupta et al., 2021; Xiao et al., 2024b; Tang et al., 2024; Cai. et al., 2024; Zhang et al., 2023; Oren et al., 2024) utilize attention sparsity to reduce the KV loading cost. KV compression can improve both latency and throughput, but suffers from accuracy degradation.

Batching has been a natural technique to improve GPU utilization by amortizing the model parameter loading cost across requests, thus boosting throughput. Recently continuous batching (Kwon et al., 2023; Yu et al., 2022; Prabhu et al., 2024) has been proposed to address the problems arising from heterogeneous batches with unequal context and generation lengths. In our work, we have considered the orthogonal direction of homogeneous batches, and the aforementioned methods are complementary to our observation.

Speculative decoding (Leviathan et al., 2022; Xia et al., 2023; Chen et al., 2023) has emerged as an algorithmic novelty to improve latency without quality degradation. SD improves latency by using a fast draft model to generate multiple tokens, which are then verified in parallel by the LLM, thus maximizing GPU utilization. However, as the batch size increases and computation resources are saturated, the verification of speculated tokens becomes costly. Hence, existing research(Liu et al., 2024a; Su et al., 2023; Miao et al., 2023; Sun et al., 2024a) has discouraged the use of speculative decoding to serve large batches of requests. In our work, we show that this claim only applies to short sequences.

To address the KV bottleneck for serving long sequences, we take inspiration from TriForce (Sun et al., 2024a), which demonstrates the effectiveness of self-speculation with compressed KV. While TriForce is designed for small batches of extremely long sequences, we have focused on large batches of moderate to long sequences, which is more nuanced in terms of draft selection. For draft selection, we have considered a subset of KV compression techniques(Xiao et al., 2024b; Li et al., 2024; Zhang et al., 2024) to exhibit the trade-off between draft cost and acceptance rate. Our work does not advocate for a single KV compression technique, rather provides a framework to choose the optimal strategy from a suite of such techniques.

Many methods have been proposed to improve speculative decoding. For instance, Speculation Parallelism (SP) (Timor et al., 2024) overlaps target verification with draft speculation to enhance speedup. This method evaluates the drafter based on draft cost and acceptance rate, which is similar to our analysis. SP complements our approach: with the high acceptance rate and low draft cost of compressed KV-based drafting, along with reduced verification costs provided by SP, speculative decoding can achieve even greater speedups in long-context serving scenarios.

## 3 THEORETICAL ANALYSIS

In this section, we present our theoretical analysis of speculative decoding and LLM inference performance. We begin by reviewing the mathematical formulation of speculative decoding speedup and identifying the key factors influencing it. Next, we analyze LLM inference in long-context scenarios, highlighting the bottleneck shift that enables speculative decoding to achieve speedup with large batch sizes. Finally, we demonstrate the necessity of compressed KV-based drafting to achieve high speedup in long-context, large batch scenarios.

### 3.1 SPECULATIVE DECODING SPEEDUP ANALYSIS

The decoding time required by the target model and the draft model for a batch of size $B$ and sequence length $S$ are given by $T_T(B,S)$ and $T_D(B,S)$ respectively. The time taken by the target model to verify

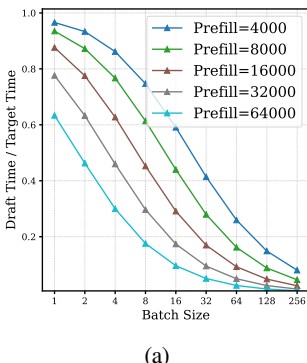 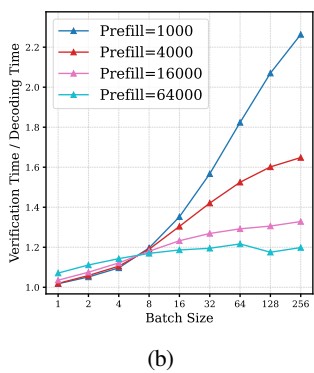 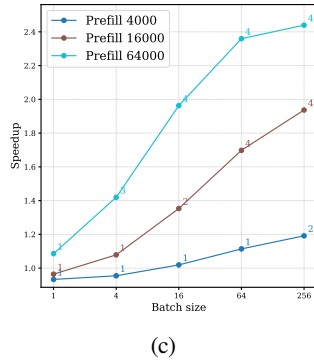

(a)            (b)            (c)

Figure 2: Theoretical analysis and expected speedup for `LLaMA-3.1-8B` deployed on 8×A100s with $\gamma = 3$. (a) Theoretical $\mathbf{T_D}/\mathbf{T_T}$ versus batch sizes. (b) Theoretical $\mathbf{T_V}(\gamma)/\mathbf{T_T}$ versus batch size. (c) Theoretical expected speedup of self-speculation across different batch sizes ( draft KV budget = 512 ).

$\gamma$ tokens is given by $T_V(B,S,\gamma)$. Given the draft token acceptance rate $\alpha \in [0,1]$ and speculation length $\gamma$, the expected number of tokens generated in one verification step is denoted by $\Omega(\gamma,\alpha)$. As described in (Leviathan et al., 2022), the expected number of generated tokens can be estimated as,

$$\Omega(\gamma,\alpha) := \mathbb{E}[\#generated\,tokens] = \frac{1-\alpha^{\gamma+1}}{1-\alpha} \tag{1}$$

The total time taken for speculative decoding, $T_{Total}^{SD}$, is given by:
$$T_{Total}^{SD} = \gamma \cdot T_D(B,S) + T_V(B,S,\gamma)$$

Hence, the expected latency per token for speculative decoding is simply $T_{Avg}^{SD} = T_{Total}^{SD}/\Omega(\gamma,\alpha)$. For brevity of notation, we will refer to these times as $T_T$, $T_D$, and $T_V$ in the future, with the dependence on $B$ and $S$ implied, unless otherwise specified.

The speedup of speculative decoding and the factors regulating it can be understood from the following equation,

$$\frac{T_{Avg}^{SD}}{T_T} = \frac{1}{\Omega(\gamma,\alpha)}\left(\frac{\gamma \cdot T_D}{T_T} + \frac{T_V(\gamma)}{T_T}\right) \tag{2}$$

From equation 2 we can see that speed-up depends on three primary factors: (a) **target verification to decoding cost ratio $\mathbf{T_V}(\gamma)/\mathbf{T_T}$**, (b) **draft to target cost ratio $\mathbf{T_D}/\mathbf{T_T}$**, and (c) **expected generation length $\Omega(\gamma,\alpha)$**. For better speedups, we aim to achieve low $T_V(\gamma)/T_T$ (close to 1), low $T_D/T_T$ (close to 0) and high $\Omega(\gamma,\alpha)$.

### 3.2 KV Cache Bottleneck Enables Speculative Decoding Speedup

In this section, we analyze how the inference bottleneck shifts as sequence length and batch size increase and how it affects the factors discussed in Section 3.1.

For short sequence lengths, speculative decoding negatively impacts batch inference efficiency (Liu et al., 2024a; Su et al., 2023). As batch size grows, the linear layers become compute-bound due to improved arithmetic intensity. This reduces the availability of compute resources that speculative decoding utilizes for parallel verification, essentially increasing the verification to decoding cost ratio.

In contrast, for moderate to long sequences, we observe a transition towards a memory-bound regime since with increasing batch size, the memory cost of loading the KV cache becomes the dominant factor. This shift from compute-bound to memory-bound inference makes the verification cost comparable to the target decoding cost. Because verification and decoding share the same KV budget, their KV cache loading costs are equivalent. The high ratio of peak FLOPS to memory bandwidth in modern GPUs causes the increase in KV loading time with batch size to outweigh the increase in computation time (see Fig. 1a). As a result, although compute-bound linear layers increase verification cost, it is mitigated by the KV bottleneck.

Based on this shift in bottlenecks, *we identify a critical sequence length $\mathbf{S_{inflection}}$, beyond which speculative decoding achieves speedup for large batches. Moreover, its speedup tends to increase with batch size.* This threshold depends on factors like the model architecture, hardware configuration, and drafting strategy.

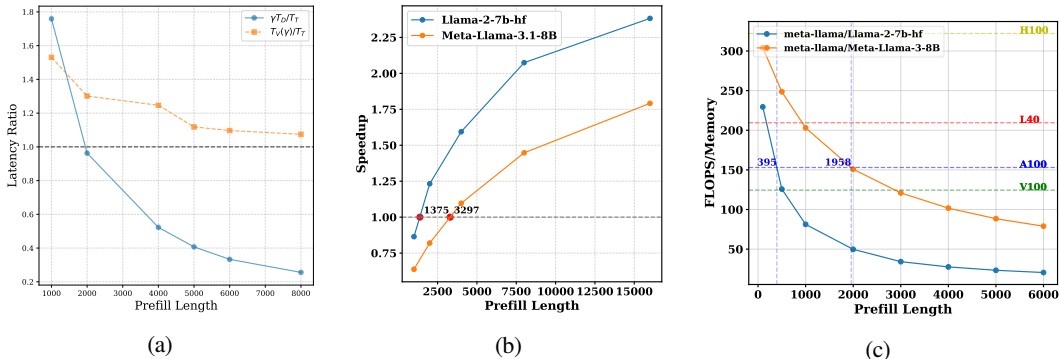

Figure 3: Theoretical analysis of self-speculation for `LLaMA-2-7B-32K` and `LLaMA-3.1-8B` with a draft KV budget of 512 and a batch size of 256. We assume the acceptance rate is 0.8 here. **(a)** Ratio of target-draft latency $(\gamma \cdot T_D / T_T)$ and verification-target latency $(T_V(\gamma)/T_T)$ versus sequence length for `LLaMA-2-7B-32K`, with $\gamma = 3$. **(b)** Theoretical speedup for different sequence lengths with a fixed $\alpha = 0.8$. **(c)** Theoretical arithmetic intensity for different sequence lengths and different models.

- For $S < S_{\text{inflection}}$:

  In this regime, as batch size increases, decoding becomes more compute-bound. Large batches can saturate the available compute, making verification relatively more expensive, as illustrated in Fig. 2b. The cost ratio $T_V(\gamma)/T_T$ increases significantly for 1000 token long sequences. If the draft token acceptance rate is low, the target model spends considerable time verifying incorrect speculations, reducing SD efficiency. Our theoretical estimate in this regime aligns with (Liu et al., 2024a). The expected speed-up with speculative decoding decreases with batch size for context lengths below the critical sequence length.

- For $S \geq S_{\text{inflection}}$:

  In this regime, speculative decoding can provide speedup for large batches, and this speedup even tends to increase with batch size when we use some intelligent drafting strategies. This happens as a combined effect of how verification to decoding cost ratio $(T_V(\gamma)/T_T)$ and draft to target cost ratio $(T_D/T_T)$ evolve with increasing batch size, as shown in Fig. 2b and 2a.

  For long sequences, KV cache loading becomes the primary bottleneck rather than compute (Sun et al., 2024a; Aminabadi et al., 2022) and the target model shifts towards memory bound regime, as shown in 3c. Because KV memory bottleneck scales with batch-size, this shift is sustained even for large batches. As the verification and decoding phases share the same KV loading cost, the cost ratio $T_V(\gamma)/T_T$ remains close to 1.

  However, the cost ratio $T_V(\gamma)/T_T$ still increases monotonically with batch size and cannot explain how we can achieve higher speedups for larger batches. The draft to target cost ratio $(T_D/T_T)$ plays an important role here. If the KV cache size of the draft model increases slower than target model, the cost ratio $T_D/T_T$ will decrease for larger batches. That is because the target model inference will be more dominated by the KV cache bottleneck rather than the draft.

As Figure 2c illustrates in the case of `LLaMA-3.1-8B`, the theoretical speedup of speculative decoding is expected to improve with increasing batch size for longer sequence lengths. The speedup decreases with batch size for $S < 4000$, but for $S \geq 4000$, the speedup increases with batch size.

As illustrated in Figure 3c, this critical sequence length $S_{\text{inflection}}$ depends on both the model's FLOPS-to-memory ratio and the GPU's FLOPS-to-memory bandwidth ratio. For a device with higher FLOPS-to-memory bandwidth ratio, we expect a lower $S_{\text{inflection}}$. Models also affect this critical sequence length. For instance, GQA model like `LLaMA-3.1-8B` tends to have higher $S_{\text{inflection}}$ due to Grouped Query Attention (GQA), which requires a larger sequence length to achieve the same KV memory footprint.

## 3.3 COMPRESSED KV CACHE ENABLES MORE EFFICIENT SPECULATION

In this section, we explain why KV compression is preferred over lightweight draft models for speculation in long-context, large batch-size scenario. There are primarily two reasons,

**KV cache grows beyond the parameter memory footprint:** Unlike parameter memory, the KV cache size grows linearly with batch size. If we use `LLaMA-3.1-8B` as a draft for `LLaMA-3.1-70B` and

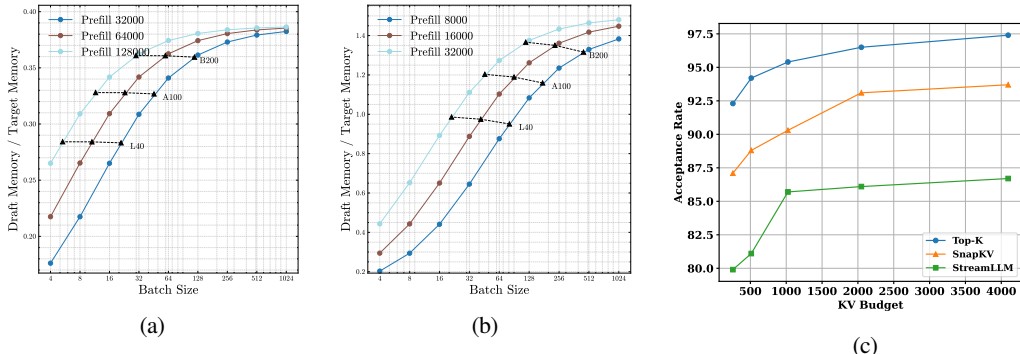

(a)  (b)  (c)

Figure 4: (a, b) Draft/target memory ratio vs batch size across different sequence lengths for `LLaMA-3.1-8B`/`LLaMA-3.1-70B` and `LLaMA-2-7B`/`LLaMA-2-70B` models. (c) `LLaMA-3.1-8B` self-speculation acceptance rate of different drafting strategy versus KV budget. Target KV length: 32000.

`LLaMA-2-7B` for `LLaMA-2-70B`, the draft models can occupy up to $38 \sim 140\%$ memory footprint of target models (Figures 4a and 4b) due to the fact that $\dim_{kv}/\dim_{model}$ is higher. Hence, in this regime, small draft models are not sufficient and compressed KV-based drafting is quite beneficial(Sun et al., 2024a). This can be seen in Figure 3a, which illustrates how $T_D/T_T$ for fixed KV size draft self-speculation with `LLaMA-3.1-8B` approaches 0 with increasing sequence length for batch size 256.

**KV compression achieves a better token acceptance rate than model compression:** A high draft token acceptance rate is critical to restrict the number of costly verification steps while serving large batches. Interestingly, we see that KV cache compression can be a more cost-effective way to improve the acceptance rate of draft tokens, especially in a high batch size long-context regime. Figure 1c illustrates this phenomenon that if a target LLM speculates itself with a sparsified version of its own KV cache, then it can achieve acceptance rates higher than those of small draft models with a full KV cache.

In summary, a draft model with compressed KV cache achieves two important factors for higher speedup in a long-context scenario: low draft cost and high acceptance rate. Figures 7b and 7c empirically illustrate the efficacy of this drafting strategy over standard SD with a small draft model in achieving higher speedups.

## 4 MAGICDEC

In this section, we present the trade-off analysis MagicDec performs to identify the correct drafting strategy. In Section 3.3, we have motivated the reason behind adopting compressed KV-based drafting in this regime. However, there are three different factors that we need to consider to effectively leverage KV compression - (a) draft model size, (b) draft KV cache size or draft KV budget, and (c) KV compression algorithm. All three factors are to be considered to strike the perfect balance between draft cost and acceptance rate.

### 4.1 GENERAL FORMULATION OF SPEEDUP WITH COMPRESSED KV-BASED DRAFTING

To begin with, we give a general formulation of speedup obtained with compressed KV-based drafting. The following analysis considers sparse KV selection algorithms; however, it can be easily extended to other KV compression methods (Hooper et al., 2024; Liu et al., 2024b; Singhania et al., 2024). The draft cost for sparse-KV methods depends on two main components: (1) draft model decoding cost, and (2) the cost of KV selection. For a given KV sparsification strategy ($select$) with a fixed KV budget of $K$, the selection cost is denoted as $T_{select}(B,S,K)$, while the decoding time for $K$ tokens is $T_D(B,K)$. The total time taken by the draft using this KV strategy with KV cache budget $K$ is:

$$T_{D,select_K}(B,S) = T_D(B,K) + T_{select}(B,S,K) \tag{3}$$

Using this as the total draft decoding time in equation 2, our final objective becomes

$$\min_{T_{select},K,\gamma,\alpha} \left[ \frac{T_{Avg}^{SD}}{T_T} \right] = \min_{T_{select},K,\gamma,\alpha} \left[ \frac{1}{\Omega(\gamma,\alpha)} \left( \frac{\gamma \cdot (T_D(B,K) + T_{select}(B,S,K))}{T_T(B,S)} + \frac{T_V(B,S,\gamma)}{T_T(B,S)} \right) \right] \tag{4}$$

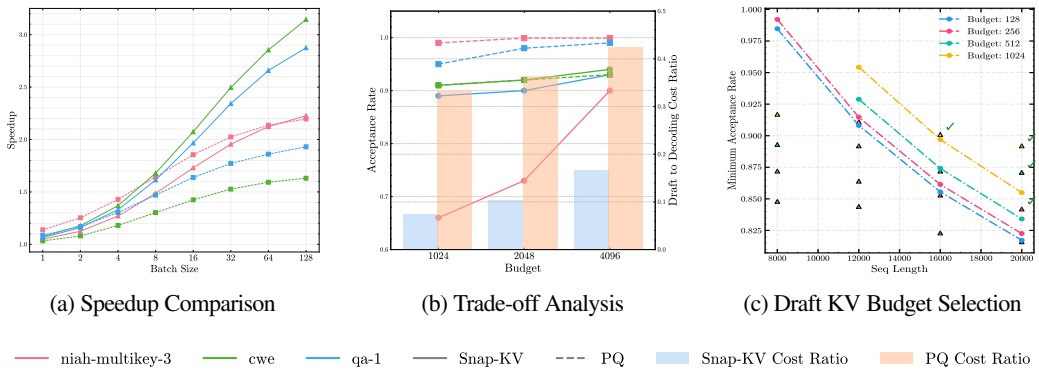

(a) Speedup Comparison  (b) Trade-off Analysis  (c) Draft KV Budget Selection

— niah-multikey-3   — cwe   — qa-1   — Snap-KV   --- PQ   ▨ Snap-KV Cost Ratio   ▨ PQ Cost Ratio

Figure 5: Comparative analysis of two KV selection algorithms - SnapKV (Li et al., 2024)(static KV selection) and PQCache (Zhang et al., 2024) (dynamic KV selection) on 3 Ruler tasks - *needle in a haystack with passkeys 3*, *common word extraction*, *question answering 1* (context length = 32,000). **(a)** Expected speed-up comparison between the two KV selection methods based on MagicDec evaluation framework. **(b)** Trade-off analysis between Draft-to-target cost ratio and acceptance rate for SnapKV and PQCache methods. **(c)** Minimum acceptance rates required to be achieved by self-speculation with different draft KV cache sizes to achieve 1.8x speedup over standard autoregressive decoding by `LLaMA-3.1-8B`. The actual acceptance rates obtained for PG-19 dataset are marked with respective colors. The admissible budgets for each sequence length are ticked right.

Now we discuss in detail the three main factors that decide the total draft decoding time $T_{D,select_K}$ and the final speedup.

## 4.2 DRAFT MODEL SIZE SELECTION

Even with a compressed KV cache, the draft model weights can play a role in deciding the best performance. The draft model parameter loading is the major part of draft cost when KV cache size is small. Usually at lower batch sizes, a small draft model with compressed KV cache can outperform self-speculation because of a lower draft to target cost ratio. When batch size and sequence length are relatively small, the parameter loading cost can impede the draft performance. Moreover, for smaller batches, the token acceptance rate requirement can be relaxed to favor a much more efficient draft model. However, beyond a certain batch size, self-speculation can become more efficient because of its higher acceptance rate, as shown in Fig. 7c.

## 4.3 DRAFT KV BUDGET SELECTION

For a fixed draft model and KV compression algorithm, the optimal draft KV cache size varies across different batch sizes and context lengths. Hence, before selecting the optimal KV compression algorithm, we need to find the respective optimal KV budgets of the candidate algorithms. We illustrate the importance of optimizing the KV budget of static KV selection algorithms for self-speculation in Figure 5c. Batches of different sequence lengths and batch sizes require different minimum acceptance rates to achieve any speedup via speculative decoding. Similarly, different KV budgets and different draft model would have different draft cost-acceptance rate trade-offs. This plot recommends the admissible draft KV budgets that reach the required minimum acceptance rate. This trade-off analysis is particularly useful for serving heterogeneous batches with different sequence lengths. Different sequences in the same batch can leverage different draft KV cache sizes to achieve the required speedup.

## 4.4 COMPARATIVE STUDY ON KV SELECTION STRATEGIES

Finally, MagicDec has to choose among different kinds of KV selection algorithms to regulate the search cost $T_{select}$. Although top-k attention can achieve very high acceptance rate with a much smaller KV cache budget, it is not a practical draft option because of its prohibitively high KV selection cost.

There are many potential alternatives to top-k attention, but determining the optimal one is not straightforward. There are primarily two kinds of KV selection algorithms - (a) dynamic KV selection algorithms such as (Tang et al., 2024; Zhang et al., 2024), (b) static KV selection algorithms such as (Xiao et al., 2024b; Yang et al., 2024; Li et al., 2024). The first kind of algorithms dynamically searches the KV cache for each input query, attempting to find the top k nearest neighbors. Although these methods can achieve higher acceptance rates, they incur substantial search costs. Conversely, static KV selection methods pre-gather a sparse KV

cache for attention approximation during generation. This approach eliminates search overhead but typically results in lower acceptance rates.

**Static vs Dynamic:** We evaluate state-of-the-art KV selection strategies using both our theoretical framework and empirical acceptance rates from self-speculation with the `LLaMA-3.1-8B` model on various Ruler tasks (Hsieh et al., 2024). Our analysis includes both static (e.g., StreamingLLM (Xiao et al., 2024b), SnapKV (Li et al., 2024)) and dynamic (e.g., PQCache (Zhang et al., 2024), TopK) KV selection algorithms, exploring different KV budgets and speculation lengths to estimate optimal theoretical speedups.

Figure 5 illustrates the trade-off between two representative KV sparsification algorithms, SnapKV and PQCache, and their respective theoretical speedups on three distinct Ruler tasks: *needle in a haystack with passkeys 3* (niah-multikeys-3), *common word extraction* (cwe), and *question answering 1* (qa-1). SnapKV[2], a static algorithm, has a lower draft-to-target cost ratio compared to PQCache, as PQCache incurs a batch-size-dependent KV selection cost $T_{select}$.

When the acceptance rates of static and dynamic methods are similar, the static method tends to dominate, as seen in the *cwe* and *qa-1* tasks. However, for the *niah-multikeys-3* task, PQCache benefits significantly from its higher acceptance rate. With an acceptance rate close to 1, PQCache can leverage longer speculation lengths, which significantly reduces the objective function in equation 4. Nevertheless, with increasing batch-size, KV search cost dominates again and the static algorithm starts to outperform the dynamic one.

## 5 EVALUATIONS

In this section, we empirically validate our theoretical analysis and demonstrate the effectiveness of our drafting strategy selection modeling. Specifically, in Section 5.1, we demonstrate the end-to-end speedup of self-speculation with sparse KV, showing that speculative decoding achieves speedup for moderate-to-long sequences, with speedup increasing as batch size grows, when sequence length exceeds a critical threshold. In Section 5.2, we compare the speedup of two drafting strategies, highlighting the effectiveness of our approach. In Section 5.3, we perform an ablation study on the speedup of speculative decoding.

### 5.1 END-TO-END SPEEDUP

We demonstrate the effectiveness of our analysis in Section 3 that speculative decoding can improve both throughput and latency for moderate-to-long sequences.

**Setup:** We use StreamingLLM (Xiao et al., 2024b) style sparse KV for drafting and conduct experiments across various batch sizes and sequence lengths to evaluate speculative decoding speedup. The system implementation details are shown in A.1. The evaluation is performed using the state-of-the-art long-context model `LLaMA-3.1-8B` on the PG-19 dataset (Rae et al., 2019). Each run generates 96 tokens per sentence in the batch through greedy decoding on 20 batches. We tested two draft KV cache budgets to assess the trade-off between draft cost and acceptance rate.

**Results:** Fig. 6 shows the speedup achieved by speculative decoding at the optimal speculation length across various batch sizes and sequence lengths. These experiments are conducted on 8xA100 GPUs.

**SD can achieve speedup for moderate to long context length.** We can find that speculative decoding consistently outperforms autoregressive decoding except when batch size is large and sequence length is short, which indicate the correctness of our analysis in Sec. 3.2.

**SD achieves better speedup with larger batch sizes.** We find that on 8xA100, when the sequence length exceeds 4000, speculative decoding achieves speedup, which increases with batch size. This result aligns with our analysis in Sec. 3.2. To verify our analysis of factors affecting the critical sequence length, we ran experiments on higher-end GPUs (H100) and lower-cost alternatives (L40), and compared the results with `LLaMA-2-7B-32K`. As shown in Table 1, the H100 achieves higher speedup than the A100 and L40 under the same setting (sequence length, batch size, and drafting strategy). This is due to the H100's higher FLOPS-to-memory bandwidth ratio, which lowers verification cost. Additionally, we can see for 8000 sequence length and the 32 batch size `LLaMA-2-7B-32K` without GQA achieves higher speedup than `LLaMA-3.1-8B` with 32000 sequence length, that's because Non-GQA model has lower FLOPS-to-memory ratio.

---

[2]SnapKV was chosen for its superior acceptance rates among static algorithms, utilizing average pooling with a kernel size of 5 and an observation window size of 32. PQCache employs product quantization with 16 sub-vectors and 8-bit quantization per key vector.

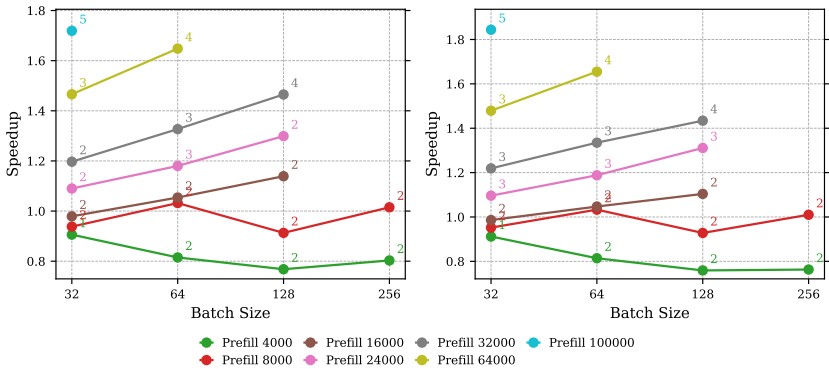

Figure 6: End-to-end speedups for StreamingLLM-based self-speculation with `LLaMA-3.1-8B` across various compressed KV budgets (left: 256, right: 512) on PG-19. Annotations indicate $\gamma_{\text{optimal}}$, which is the value corresponding to the highest speedup achieved. Experiments are conducted on 8xA100 with 8-way tensor parallelism. Raw data can be found in A.2.

Table 1: Results on L40 and H100, StreamingLLM budget for the draft model is 512, each with the optimal $\gamma$

| Target | Draft | Task | GPU | Prefill | Bsz | $\gamma$ | $\gamma T_D(1)$ | $T_V(\gamma)$ | $\Omega(\gamma,\alpha)$ | $T^{AR}$ | $T^{SD}$ | x |
|--------|-------|------|-----|---------|-----|----------|-----------------|---------------|-------------------------|----------|----------|---|
| Llama3.1-8B | StreamingLLM | PG-19 | 8xL40 | 32000 | 32 | 3 | 44.11 | 45.12 | 3.00 | 36.62 | 30.32 | 1.21 |
| Llama2-7B-32K | StreamingLLM | PG-19 | 8xL40 | 8000 | 32 | 2 | 29.06 | 42.02 | 2.53 | 35.13 | 28.70 | 1.22 |
| Llama2-7B-32K | StreamingLLM | PG-19 | 8xL40 | 8000 | 64 | 3 | 58.33 | 74.85 | 3.14 | 62.92 | 42.96 | 1.46 |
| Llama3.1-8B | StreamingLLM | PG-19 | 4xH100 | 32000 | 32 | 3 | 15.09 | 18.30 | 2.82 | 17.32 | 12.16 | 1.42 |
| Llama2-7B-32K | StreamingLLM | PG-19 | 4xH100 | 8000 | 32 | 3 | 14.20 | 15.64 | 2.98 | 14.85 | 10.29 | 1.44 |
| Llama2-7B-32K | StreamingLLM | PG-19 | 4xH100 | 8000 | 64 | 4 | 23.63 | 27.90 | 3.37 | 26.17 | 15.58 | 1.68 |

## 5.2 COMPARING DIFFERENT KV COMPRESSION METHODS

In this section, we compare two static KV compression methods for drafting, with results shown Fig. 7b and Fig. 7c. The detail results are in Table 6. We perform a sweep to select the optimal speculation length and KV budget for each method. The best draft budget for StreamingLLM-based self-speculation is 512, while for SnapKV-based approach, it is 2049. The results indicate that SnapKV-based drafting outperforms StreamingLLM for self-speculation in all the cases. Based on Fig. 4c and our analysis in Sec. 4, the key factor is the acceptance rate. Both StreamingLLM and SnapKV are static KV compression methods, so neither incurs KV search overhead. However, SnapKV has a much higher acceptance rate, which increases rapidly with KV budget, mitigating the rise in draft cost. In contrast, StreamingLLM's acceptance rate has a lower upper bound and increases more slowly with KV budget. As a result, SnapKV achieves higher speedup due to the combined effect of acceptance rate and draft cost. We further evaluated SnapKV-based self-speculation across different batch sizes, sequence lengths, and tasks, with promising results. As shown in Table 2, SnapKV-based self-speculation achieves up to **2.51x** speedup, demonstrating speculative decoding's ability to improve throughput.

Table 2: Further Results of SnapKV Self-speculation on Different Tasks

| Target | Draft | Task | GPU | Prefill | Bsz | $\gamma$ | $\gamma T_D(1)$ | $T_V(\gamma)$ | $\Omega(\gamma,\alpha)$ | $T^{AR}$ | $T^{SD}$ | x |
|--------|-------|------|-----|---------|-----|----------|-----------------|---------------|-------------------------|----------|----------|---|
| Llama3.1-8B | SnapKV | PG-19 | 8xH100 | 100000 | 41 | 7 | 34.34 | 28.50 | 5.61 | 25.96 | 11.35 | 2.29 |
| Llama3.1-8B | SnapKV | QA-1 | 8xH100 | 100000 | 41 | 11 | 53.90 | 29.89 | 7.93 | 25.90 | 10.64 | 2.43 |
| Llama3.1-8B | SnapKV | CWE | 8xH100 | 100000 | 41 | 11 | 53.98 | 29.93 | 8.21 | 25.83 | 10.29 | 2.51 |
| Llama3.1-8B | SnapKV | PG-19 | 8xH100 | 64000 | 64 | 6 | 32.89 | 28.80 | 5.41 | 25.52 | 11.54 | 2.21 |
| Llama3.1-8B | SnapKV | QA-1 | 8xH100 | 64000 | 64 | 7 | 38.40 | 29.11 | 6.08 | 25.43 | 11.20 | 2.27 |
| Llama3.1-8B | SnapKV | CWE | 8xH100 | 64000 | 64 | 8 | 43.91 | 29.29 | 6.83 | 25.48 | 10.81 | 2.36 |

## 5.3 ABLATION STUDY

In this section, we present ablation studies of our speculative decoding speedup analysis model.

**Draft KV Budget.** As modeled in Section 4, the selection of KV budget depends on verification cost, acceptance rate, and draft cost. As shown in Fig. 6, when batch size and sequence length are large, a larger KV budget results in higher speedup. In this scenario, the LLM is highly memory-bound, so verification

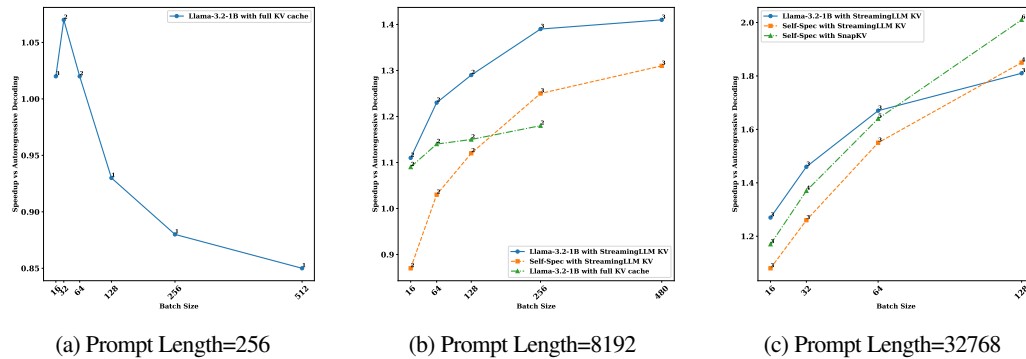

(a) Prompt Length=256        (b) Prompt Length=8192        (c) Prompt Length=32768

Figure 7: Comparison between different drafting strategy for `LLaMA-3.1-8B` under short, medium and long context length across batch sizes. Hardware: 8xH100. Each with optimal gamma. Dataset: PG-19.

cost is low, but its absolute value is much larger than the draft cost with a fixed KV size. Therefore, a larger KV budget with a higher acceptance rate is preferred to increase the average generation length per step.

**Draft Model Weights.** Draft model weights loading is also a part of draft cost. We have several choices of drafting stategy with the trade-off of draft cost and acceptance rate. A small draft model can have much lower model weights loading cost, but with significant lower acceptance rate. We conduct experiments under prompt length 256, 8192 and 32768 to show the effect to speedup of different draft model selection. The results are shown in Fig. 7. We can see in Fig. 7b that when sequence length is not sufficient long and batch size is not very large, small draft model with the KV compression tends to outperform self-speculation. This is because, in these scenarios, KV doesn't fully dominate inference, and model weight loading makes draft costs of self-speculation a lot higher. However, when both sequence length and batch size are very large, and the KV cache dominates LLM inference, self-speculation surpasses the small draft model, as model weight loading contributes minimally to overall latency. The high acceptance rate of compressed KV self-speculation has higher speedup upper bound, and leads to better speedup when batch size is large, as demonstrated in Fig. 7c.

**Models.** Different models have different FLOPS to Memory Ratio and acceptance rate. We also conducted experiments on `Qwen2.5-7B` , `Qwen2.5-32B` and `Mistral-7B-v0.3` models to show the generalizability of MagicDec. The results are shown in Sec. A.5. We can see speculative decoding works well for these models, achieving up to 2.06x speedup for `Mistral-7B-v0.3` , 1.89x speedup for `Qwen2.5-7B` and 1.51x speedup for `Qwen2.5-32B` on PG-19 dataset. The trend of speedup also matches our previous analysis and the `LLaMA-3.1-8B` results.

## 6   CONCLUSION AND LIMITATION

Optimizing both throughput and latency for LLM inference is challenging, especially for long-context, large batch-size regime. Our analysis reveals that speculative decoding can be beneficial in this regime, with its efficacy increasing with larger batch-sizes, contrary to existing misconceptions. In search of effective drafting strategies, we discover that KV compression is easier than model compression to achieve higher acceptance rate at the same memory budget, which becomes more prominent in high batch-size and long context-length regime. Leveraging these insights, we explore different KV compression algorithms for drafting and present a bottleneck-aware general formulation to select suitable drafting strategy based on task, batch-size and sequence-length. MagicDec only focuses on decoding performance for long-context LLM serving, while the prefill is also very challenging in this scenario. There has been some work focusing on improving the prefill performance (Agrawal et al., 2024a; Zhong et al., 2024), which could be integerated with MagicDec to improve both prefill and decode performance. MagicDec tends to achieve better speedup on high-end GPUs due to their higher FLOPS-to-memory bandwidth ratio and large HBM size. Future work can explore the adoption of speculative decoding on offloading and distributed setting to reduce the communication overhead, thus better utilize the resource of commodity devices.

## 7   ACKNOWLEDGEMENTS

We would like to thank Xinyu Yang, Yang Zhou, Harry Dong, Haizhong Zheng, Hanshi Sun, and the anonymous reviewers for providing us constructive feedback on our paper. This work was partially supported by Li Auto, Together AI and Moffett AI.

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

## A APPENDIX

### A.1 SYSTEM IMPLEMENTATION

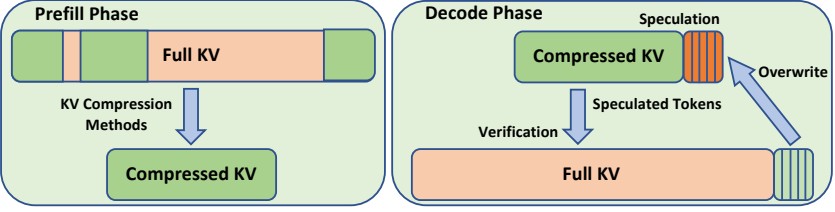

Figure 8: Self-Speculation System Design. We demonstrate using a static KV compression method.

The design of our speculative decoding system is shown in Fig. 8, demonstrating the use of a static KV compression method. The static compressed KV is generated during prefill phase and used for drafting. We implement the speculative decoding system on both state-of-the-art inference framework MLC-LLM (team, 2023) and a self-implement inference backend. The main results are obatined from our self-implemented backend. The comparison of our backend and MLC-LLM can be found in A.3.

The self-implement inference backend is built on GPT-Fast (pytorch-labs, 2023), with Flashinfer (flashinfer-ai) accelerating attention computation. We use torch.compile to compile the model and utilize Triton-based matrix multiplication to accelerate the MLP layers. We use Pytorch CUDA graphs to reduce CPU kernel launch overhead. These optimizations help minimize overhead and improve speedup. We also implement tensor parallelism for the embedding layer to further accelerate drafting.

## A.2   RESULTS OF VARIOUS BATCH SIZE AND CONTEXT LENGTH ON A100

We show the raw data points we collected when running speculative decoding on the self-implement backend to support our previous discussion. We sweep the batch size and sequence lengths, and compare the speedup of different drafting strategy for different models. We ran all these experiments on 8 Nvidia A100 GPU with 8-way Tensor Parallelism.

(a) `LLaMA-2-7B-32K` , `TinyLLama-1.1B`

| S | B | $\gamma T_D$ | $T_V$ | $\Omega$ | $T^{AR}$ | $T^{SD}$ | x |
|---|---|---|---|---|---|---|---|
| 1024 | 32 | 8.21 | 9.55 | 2.19 | 8.27 | 8.70 | 0.95 |
| 1024 | 48 | 8.46 | 10.66 | 2.19 | 9.41 | 9.33 | 1.01 |
| 1024 | 64 | 9.26 | 13.05 | 2.19 | 10.83 | 10.80 | 1.00 |
| 1024 | 128 | 12.04 | 18.87 | 2.19 | 14.02 | 14.83 | 0.94 |
| 4000 | 32 | 8.46 | 13.21 | 2.19 | 11.89 | 10.52 | 1.13 |
| 4000 | 48 | 8.71 | 16.19 | 2.19 | 14.39 | 12.02 | 1.20 |
| 4000 | 64 | 9.35 | 21.83 | 2.19 | 19.28 | 14.88 | 1.30 |
| 4000 | 128 | 12.31 | 33.82 | 2.19 | 28.77 | 21.78 | 1.32 |
| 8000 | 32 | 8.61 | 18.40 | 2.18 | 16.53 | 13.02 | 1.27 |
| 8000 | 48 | 8.91 | 23.67 | 2.18 | 21.45 | 15.58 | 1.38 |
| 8000 | 64 | 9.58 | 34.32 | 2.18 | 31.49 | 20.80 | 1.51 |
| 8000 | 128 | 12.54 | 53.78 | 2.18 | 49.89 | 31.25 | 1.60 |
| 16000 | 32 | 8.78 | 27.79 | 2.17 | 26.28 | 17.46 | 1.50 |
| 16000 | 48 | 9.33 | 38.29 | 2.18 | 35.83 | 22.52 | 1.59 |
| 16000 | 64 | 9.92 | 58.14 | 2.17 | 55.08 | 31.99 | 1.72 |
| 24000 | 32 | 8.68 | 37.57 | 2.16 | 35.70 | 22.05 | 1.62 |
| 32000 | 32 | 8.83 | 47.35 | 2.17 | 44.94 | 26.55 | 1.69 |

(b) `LLaMA-2-7B-32K` Self Speculation

| S | B | $\gamma T_D$ | $T_V$ | $\Omega$ | $T^{AR}$ | $T^{SD}$ | x |
|---|---|---|---|---|---|---|---|
| 4000 | 32 | 15.42 | 13.17 | 2.56 | 11.89 | 11.69 | 1.02 |
| 4000 | 48 | 16.96 | 16.38 | 2.56 | 14.39 | 13.55 | 1.06 |
| 4000 | 64 | 19.75 | 22.01 | 2.57 | 19.28 | 16.82 | 1.15 |
| 4000 | 128 | 25.82 | 33.79 | 2.56 | 28.77 | 23.86 | 1.21 |
| 8000 | 32 | 15.70 | 18.23 | 2.53 | 16.53 | 13.99 | 1.18 |
| 8000 | 48 | 18.44 | 24.32 | 2.53 | 21.45 | 17.50 | 1.23 |
| 8000 | 64 | 20.03 | 34.30 | 2.53 | 31.49 | 22.05 | 1.43 |
| 8000 | 128 | 26.10 | 53.69 | 2.52 | 49.89 | 32.25 | 1.55 |
| 16000 | 32 | 16.06 | 27.54 | 2.50 | 26.28 | 18.02 | 1.46 |
| 16000 | 48 | 19.75 | 39.03 | 2.50 | 35.83 | 24.15 | 1.48 |
| 16000 | 64 | 20.87 | 58.15 | 2.51 | 55.08 | 32.16 | 1.71 |
| 24000 | 32 | 15.80 | 37.06 | 2.49 | 35.70 | 21.77 | 1.64 |
| 32000 | 32 | 16.19 | 46.55 | 2.50 | 44.94 | 25.64 | 1.75 |

(c) `LLaMA-3.1-8B` Self Speculation

| S | B | $\gamma T_D$ | $T_V$ | $\Omega$ | $T^{AR}$ | $T^{SD}$ | x |
|---|---|---|---|---|---|---|---|
| 4000 | 32 | 13.16 | 10.32 | 2.54 | 8.83 | 9.78 | 0.90 |
| 4000 | 64 | 16.48 | 13.55 | 2.54 | 10.07 | 12.36 | 0.81 |
| 4000 | 128 | 23.41 | 19.77 | 2.54 | 13.42 | 17.70 | 0.76 |
| 4000 | 256 | 39.29 | 35.05 | 2.53 | 23.23 | 30.46 | 0.76 |
| 8000 | 32 | 13.28 | 11.34 | 2.50 | 9.90 | 10.40 | 0.95 |
| 8000 | 64 | 16.98 | 16.06 | 2.51 | 14.16 | 13.72 | 1.03 |
| 8000 | 128 | 23.59 | 24.84 | 2.51 | 18.53 | 19.97 | 0.93 |
| 8000 | 256 | 39.32 | 46.44 | 2.51 | 35.35 | 34.99 | 1.01 |
| 16000 | 32 | 14.46 | 14.00 | 2.47 | 11.93 | 12.10 | 0.99 |
| 16000 | 64 | 18.00 | 21.15 | 2.48 | 17.17 | 16.40 | 1.05 |
| 16000 | 128 | 25.77 | 34.82 | 2.46 | 28.00 | 25.36 | 1.10 |
| 32000 | 32 | 14.12 | 19.04 | 2.46 | 17.13 | 14.05 | 1.22 |
| 32000 | 64 | 19.08 | 30.86 | 2.45 | 26.99 | 21.03 | 1.28 |
| 32000 | 128 | 28.26 | 54.98 | 2.45 | 47.24 | 34.94 | 1.35 |
| 64000 | 32 | 14.92 | 28.88 | 2.40 | 26.96 | 18.91 | 1.43 |
| 64000 | 64 | 18.25 | 50.19 | 2.40 | 46.09 | 29.22 | 1.58 |
| 100000 | 32 | 15.10 | 39.84 | 2.45 | 37.70 | 23.05 | 1.64 |

Table 3: Comparison of results for different LLaMA models and configurations (budget=512 and $\gamma = 2, 8 \times$ A100). Here S and B represent prefill length and batch size, respectively.

## A.3   COMPARISON WITH MLC-LLM RESULTS

We compare the results of SnapKV based self-speculation on MLC-LLM and our backend. As the measurement methods are different, we put them in two tables as shown in Table 4 and 5. The verification time of MLC-LLM includes one step of draft decode time. Our backend is highly optimized for speculative decoding setting, minimizing the drafting and verification overhead, thus, leading to better speedup. However, the trend that speedup increases with batch size is the same, aligning with our theoretical analysis in Section 3.

Table 4: Results of Our Backend

| Target | Backend | Task | GPU | Prefill | Bsz | $\gamma$ | $\gamma T_D(1)$ | $T_V(\gamma)$ | $\Omega(\gamma, \alpha)$ | $T^{AR}$ | $T^{SD}$ | x |
|---|---|---|---|---|---|---|---|---|---|---|---|---|
| Llama3.1-8B | Ours | PG-19 | 8xH100 | 32000 | 16 | 3 | 10.96 | 6.91 | 3.42 | 6.41 | 5.41 | 1.18 |
| Llama3.1-8B | Ours | PG-19 | 8xH100 | 32000 | 32 | 4 | 16.69 | 10.39 | 4.10 | 9.23 | 6.75 | 1.37 |
| Llama3.1-8B | Ours | PG-19 | 8xH100 | 32000 | 64 | 5 | 23.96 | 17.45 | 4.59 | 14.85 | 9.17 | 1.62 |

Table 5: Results of MLC-LLM

| Target | Backend | Task | GPU | Prefill | Bsz | $\gamma$ | $T_D(1)$ | $T_V(\gamma)$ | NumGen | ARTrput | SDTrput | x |
|---|---|---|---|---|---|---|---|---|---|---|---|---|
| Llama3.1-8B | MLC-LLM | PG-19 | 8xH100 | 32000 | 16 | 4 | 3.64 | 13.60 | 724 | 2471.4 | 2133.0 | 0.86 |
| Llama3.1-8B | MLC-LLM | PG-19 | 8xH100 | 32000 | 32 | 4 | 4.19 | 16.13 | 1455 | 3311.5 | 3664.5 | 1.11 |
| Llama3.1-8B | MLC-LLM | PG-19 | 8xH100 | 32000 | 64 | 5 | 5.27 | 28.26 | 2719 | 3930.0 | 4959.2 | 1.26 |

## A.4 FURTHER SNAPKV AND STREAMINGLLM RESULTS

We show the raw experiment data. We compare both StreamingLLM-based self-speculation and SnapKV-based self-speculation, and also a small draft model with StreamingLLM KV cache.

Table 6: Comparison of SnapKV, StreamingLLM, and Tiny Draft (StreamingLLM KV) Speculation. Each with optimal $\gamma$ and KV budget

| Target | Draft | Task | GPU | Prefill | Bsz | $\gamma$ | $\gamma T_D(1)$ | $T_V(\gamma)$ | $\Omega(\gamma,\alpha)$ | $T^{AR}$ | $T^{SD}$ | x |
|---|---|---|---|---|---|---|---|---|---|---|---|---|
| Llama3.1-8B | Llama3.2-1B(S) | PG-19 | 8xH100 | 32000 | 16 | 3 | 4.43 | 6.71 | 2.43 | 6.18 | 4.86 | 1.27 |
| Llama3.1-8B | StreamingLLM | PG-19 | 8xH100 | 32000 | 16 | 3 | 10.33 | 6.73 | 3.09 | 6.18 | 5.74 | 1.08 |
| Llama3.1-8B | SnapKV | PG-19 | 8xH100 | 32000 | 16 | 3 | 10.55 | 6.84 | 3.41 | 6.18 | 5.27 | 1.17 |
| Llama3.1-8B | Llama3.2-1B(S) | PG-19 | 8xH100 | 32000 | 32 | 3 | 4.71 | 9.70 | 2.43 | 9.10 | 6.22 | 1.46 |
| Llama3.1-8B | StreamingLLM | PG-19 | 8xH100 | 32000 | 32 | 3 | 11.55 | 9.74 | 3.06 | 9.10 | 7.20 | 1.26 |
| Llama3.1-8B | SnapKV | PG-19 | 8xH100 | 32000 | 32 | 4 | 15.79 | 10.36 | 4.03 | 9.10 | 6.64 | 1.37 |
| Llama3.1-8B | Llama3.2-1B(S) | PG-19 | 8xH100 | 32000 | 64 | 3 | 5.05 | 15.86 | 2.44 | 14.84 | 8.88 | 1.67 |
| Llama3.1-8B | StreamingLLM | PG-19 | 8xH100 | 32000 | 64 | 3 | 12.82 | 15.93 | 3.08 | 14.84 | 9.57 | 1.55 |
| Llama3.1-8B | SnapKV | PG-19 | 8xH100 | 32000 | 64 | 5 | 22.91 | 17.70 | 4.55 | 14.84 | 9.05 | 1.64 |
| Llama3.1-8B | Llama3.2-1B(S) | PG-19 | 8xH100 | 32000 | 128 | 3 | 5.79 | 28.51 | 2.43 | 26.07 | 14.43 | 1.81 |
| Llama3.1-8B | StreamingLLM | PG-19 | 8xH100 | 32000 | 128 | 4 | 18.96 | 30.34 | 3.57 | 26.07 | 14.06 | 1.85 |
| Llama3.1-8B | SnapKV | PG-19 | 8xH100 | 32000 | 128 | 6 | 33.33 | 31.60 | 5.07 | 26.07 | 12.96 | 2.01 |

## A.5 RESULTS OF QWEN AND MISTRAL MODELS

Table 7: Results of Qwen and Mistral Models. Each with optimal $\gamma$ and KV budget

| Target | Draft | Task | GPU | Prefill | Bsz | $\gamma$ | $\gamma T_D(1)$ | $T_V(\gamma)$ | $\Omega(\gamma,\alpha)$ | $T^{AR}$ | $T^{SD}$ | x |
|---|---|---|---|---|---|---|---|---|---|---|---|---|
| Mistral-7B-v0.3 | SnapKV | PG-19 | 8xH100 | 32000 | 32 | 3 | 11.71 | 9.62 | 3.49 | 8.92 | 6.12 | 1.46 |
| Mistral-7B-v0.3 | SnapKV | PG-19 | 8xH100 | 32000 | 64 | 3 | 13.64 | 15.64 | 3.47 | 14.49 | 8.44 | 1.72 |
| Mistral-7B-v0.3 | SnapKV | PG-19 | 8xH100 | 32000 | 128 | 5 | 27.49 | 30.65 | 4.72 | 25.41 | 12.31 | 2.06 |
| Qwen-2.5-7B | SnapKV | PG-19 | 4xH100 | 32000 | 32 | 3 | 11.40 | 9.26 | 3.40 | 8.20 | 6.07 | 1.35 |
| Qwen-2.5-7B | SnapKV | PG-19 | 4xH100 | 32000 | 64 | 4 | 17.67 | 15.67 | 4.06 | 13.11 | 8.20 | 1.6 |
| Qwen-2.5-7B | SnapKV | PG-19 | 4xH100 | 32000 | 128 | 5 | 27.22 | 28.51 | 4.62 | 22.79 | 12.06 | 1.89 |
| Qwen-2.5-32B | SnapKV | PG-19 | 8xH100 | 32000 | 8 | 3 | 23.67 | 11.98 | 3.50 | 10.42 | 10.19 | 1.02 |
| Qwen-2.5-32B | SnapKV | PG-19 | 8xH100 | 32000 | 16 | 3 | 25.27 | 15.29 | 3.52 | 13.36 | 11.52 | 1.16 |
| Qwen-2.5-32B | SnapKV | PG-19 | 8xH100 | 32000 | 32 | 3 | 28.99 | 21.90 | 3.51 | 19.43 | 14.49 | 1.34 |
| Qwen-2.5-32B | Qwen-2.5-7B | PG-19 | 8xH100 | 32000 | 8 | 2 | 9.04 | 11.31 | 2.32 | 10.42 | 8.74 | 1.19 |
| Qwen-2.5-32B | Qwen-2.5-7B | PG-19 | 8xH100 | 32000 | 16 | 2 | 11.61 | 14.59 | 2.32 | 13.36 | 11.31 | 1.18 |
| Qwen-2.5-32B | Qwen-2.5-7B | PG-19 | 8xH100 | 32000 | 32 | 2 | 16.72 | 20.87 | 2.31 | 19.43 | 16.27 | 1.19 |
| Qwen-2.5-32B | Qwen-2.5-7B(Streaming) | PG-19 | 8xH100 | 32000 | 8 | 2 | 6.77 | 11.31 | 2.27 | 10.42 | 7.97 | 1.31 |
| Qwen-2.5-32B | Qwen-2.5-7B(Streaming) | PG-19 | 8xH100 | 32000 | 16 | 2 | 7.21 | 14.59 | 2.26 | 13.36 | 9.64 | 1.39 |
| Qwen-2.5-32B | Qwen-2.5-7B(Streaming) | PG-19 | 8xH100 | 32000 | 32 | 3 | 11.78 | 21.82 | 2.62 | 19.43 | 12.85 | 1.51 |

## A.6 TINYLLAMA1.1B-LLAMA2-7B-32K RESULTS

We also test the non-GQA model `LLaMA-2-7B-32K` for both StreamingLLM-based self-speculation and small draft model with StreamingLLM KV cache. Due to the lower FLOPS to memory ratio of non-GQA model, it tends to achieve higher speedup than GQA model under the same setting.

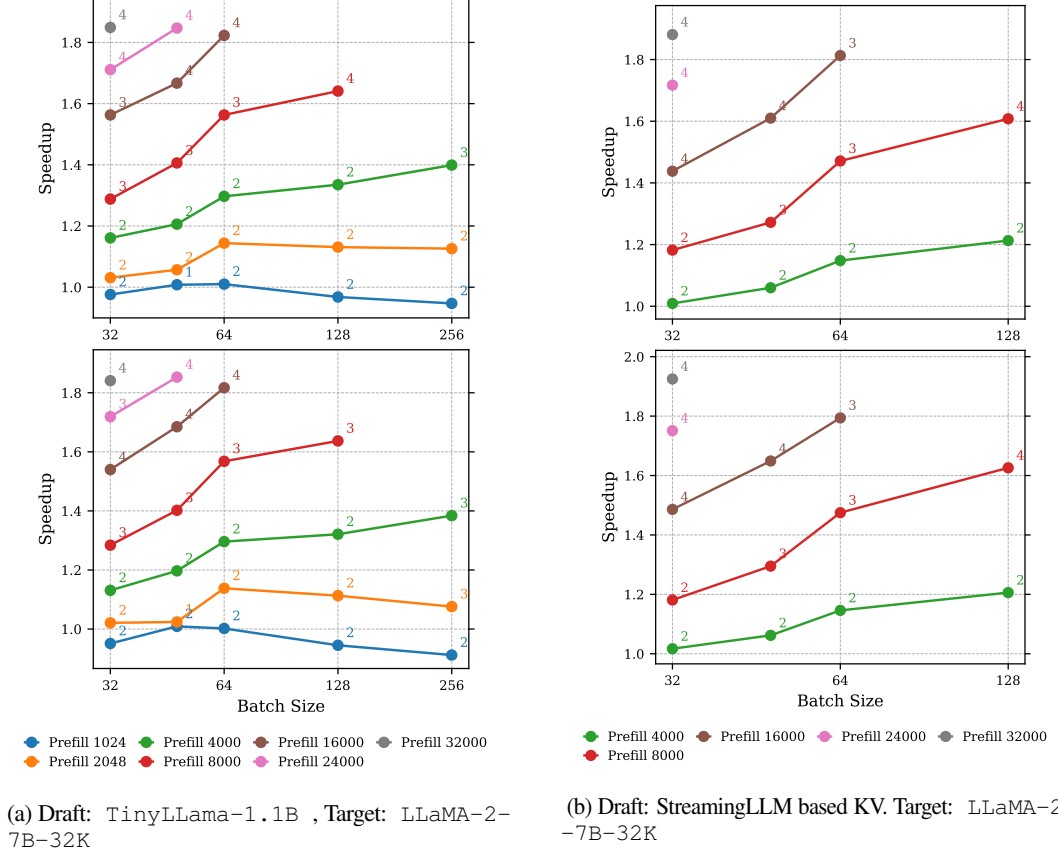

(a) Draft: `TinyLLama-1.1B`, Target: `LLaMA-2-7B-32K`

(b) Draft: StreamingLLM based KV. Target: `LLaMA-2-7B-32K`

Figure 9: End-to-end speedups for StreamingLLM-based self-speculation across various compressed KV budgets (left: 256, right: 512) on PG-19. Annotations indicate $\gamma_{\text{optimal}}$, which is the value corresponding to the highest speedup achieved. Experiments are conducted on 8xA100 with 8-way tensor parallelism. Raw data can be found in A.2.

