# OpenReview forum: "MagicDec: Breaking the Latency-Throughput Tradeoff for Long Context Generation with Speculative Decoding"
_ICLR.cc/2025/Conference — ICLR 2025 Poster_

### Official Review · Reviewer_3wSP · 2024-10-29

**Soundness:** 3
**Presentation:** 3
**Contribution:** 2
**Rating:** 8
**Confidence:** 2

**Summary:**

Conventional understanding suggests that **speculative decoding (SD)** enhances performance primarily in scenarios with small batch sizes. This paper presents a novel theoretical analysis demonstrating that SD can also yield performance improvements in settings with large batch sizes and extended prompt lengths. The analysis in this paper identifies how performance bottlenecks shift with increasing batch size and prompt length. To address these bottlenecks, this paper proposes using a draft model with a compressed key-value (KV) cache, effectively alleviating the new constraints. The theoretical framework provided by this paper enables an optimal drafting strategy tailored to specific draft-target model pairs, making this approach particularly valuable for SD applications in long-context scenarios.

**Strengths:**

1.The paper is well-written.
2. The paper presents a clear and logical progression of ideas.
3. The paper provides rigorous theoretical analysis supported by extensive experimental results.

**Weaknesses:**

I appreciate the quality of this paper, and I have only one minor suggestion.

1. **Potential limitations in real-world application**: The theoretical analysis is thorough and effectively clarifies when the conventional view—that speculative decoding (SD) enhances performance primarily with small batch sizes—applies, as well as when the new discovery on SD’s performance benefits with larger batch sizes and extended prompt lengths holds true. However, in real-world cloud environments, request configurations are often limited to smaller batch sizes and shorter prompt lengths (see https://www.microsoft.com/en-us/research/publication/splitwise-efficient-generative-llm-inference-using-phase-splitting/). Could the authors kindly discuss how this theoretical analysis might be applied within the constraints of typical LLM request configurations?

**Questions:**

Please answer the following question.

1. **Potential limitations in real-world application**: The theoretical analysis is thorough and effectively clarifies when the conventional view—that speculative decoding (SD) enhances performance primarily with small batch sizes—applies, as well as when the new discovery on SD’s performance benefits with larger batch sizes and extended prompt lengths holds true. However, in real-world cloud environments, request configurations are often limited to smaller batch sizes and shorter prompt lengths (see https://www.microsoft.com/en-us/research/publication/splitwise-efficient-generative-llm-inference-using-phase-splitting/). Could the authors kindly discuss how this theoretical analysis might be applied within the constraints of typical LLM request configurations?

---

> ### Author Response · Authors · 2024-11-21
> **Response to Reviewer 3wSP**
>
> We thank the reviewer for the supportive feedback. We are glad that you appreciated our work. And we have run some additional experiments, combined with theoretical analysis, to address your concern of real-world application of MagicDec.
>
> ---
>
> ### **Q1. Potential limitations in real-world application**
>
> We thank the reviewer for highlighting the practical constraints of real-world configurations in cloud environments. We agree that very long prompts in large batches perhaps remain uncommon in typical LLM usage patterns. However, MagicDec can also get gain for moderate length input prompt and medium batch size without adding any cost or hurting accuracy. As shown in Figure 3 (Page 5) of our paper, for 8xA100 GPU with a batch size of 256, speculative decoding becomes beneficial once the context length exceeds 3297 tokens, a threshold we term the **critical sequence length**. For modern GPUs in cloud centers like the H100, this critical length is even lower due to the higher FLOPs-to-memory bandwidth ratio. This critical prompt length is not a very large value and is more common in real-world applications.
>
> To demonstrate practical applicability, we tested MagicDec using Llama-3.2-1B with streamingLLM KV for speculation of Llama-3.1-8B. The draft budget is 256. We used a prompt length of 3072 and a batch size of 128. Results are shown below:
>
> | GPU    | Bsz | Prompt Len | Gamma | T_drf | T_ver | acc_len | T_sd  | T_auto | Speedup |
> |--------|-----|------------|-------|-------|-------|---------|-------|--------|---------|
> | 8xH100 | 128 | 3072       | 1     | 2.03  | 6.97  | 1.70    | 5.73  | 6.27   | 1.09    |
>
> For this moderate sequence length and batch size, MagicDec demonstrates speedup over standard decoding. As batch size or sequence length increases, the speedup becomes even more pronounced. The growth of sequence length is the current trend.
>
> Recently, with the emergence of long-context models and applications like retrieval-augmented generation and in-context learning, the input context lengths have increased significantly in real-world serving systems. Production systems like Anthropic have reported this surge in context length of the input prompts [1]. Moreover, they often augment the user prompts with extra context to generate better responses. For instance, OpenAI o1 has built-in reasoning capabilities and it usually augments user prompts internally with chain-of-thought prompting. Hence, we believe that MagicDec will become more and more relevant in coming years.
>
> **[1]** Anthropic: Prompt Caching with Claude. https://www.anthropic.com/news/prompt-caching

---

> > ### Comment · Reviewer_3wSP · 2024-11-26
> >
> > The authors have answered my questions. I increase my score.

---

> ### Author Response · Authors · 2024-11-25
>
> Dear Reviewer 3wSP,
>
> Thank you for your thoughtful and constructive feedback on our work. We have carefully addressed your comments and revised the work and manuscript accordingly. As the discussion period nears its end, we would greatly appreciate any additional questions or points of clarification. If our responses have satisfactorily addressed your concerns, we kindly ask you to consider reflecting this in your score.
>
> Thanks again for your time and expertise.

---

### Official Review · Reviewer_UdZh · 2024-11-03

**Soundness:** 3
**Presentation:** 3
**Contribution:** 4
**Rating:** 8
**Confidence:** 4

**Summary:**

The paper addresses the challenge of improving latency and throughput in LLM inference for long-context tasks. Traditional speculative decoding (SD) literature tends to focus on smaller batch sizes, but MagicDec demonstrates how SD can also benefit high-throughput scenarios involving long sequences. By combining theoretical and empirical approaches, MagicDec identifies and mitigates bottlenecks in memory access using a draft model with sparse Key-Value (KV) caching. It introduces a theoretical model for selecting optimal draft strategies, achieving speedups of up to 2.51x in large batch settings for the LLaMA-3.1-8B model across diverse hardware configurations.

**Strengths:**

+ The work introduces novel improvements in speculative decoding for large-batch, long-sequence LLMs, challenging existing assumptions and offering a fresh perspective.
+ Theoretical insights are well-validated through high-quality experimental results, with robust data supporting the effectiveness of MagicDec across diverse scenarios.
+ While some sections could benefit from increased clarity, the paper's main findings are well-articulated, with sufficient depth to support the claims.
+ MagicDec offers notable improvements in both latency and throughput for LLMs, with potential impacts across a range of long-context applications.

**Weaknesses:**

- Complexity in Explanation: Sections discussing KV caching and speculative decoding speedup factors could be streamlined to improve readability, especially for a broader audience.

**Questions:**

## Questions
1. Can you clarify the potential trade-offs in performance if MagicDec were applied to significantly smaller LLM models?
2. Is the performance gain from MagicDec sustained across varying types of long-context tasks, particularly those that require variable batch sizes or non-standard hardware configurations?
3. Have you considered additional draft model selection criteria that might further alleviate the KV bottleneck?

## Comments
### Comment on Lines 52-74 and Figure 1a
In the section describing how the "KV Cache Bottleneck Enables SD Speedup Even For Large Batches," the authors assert that KV cache loading time increases significantly in long-context, large-batch scenarios, leading to a more memory-bound process. This serves as evidence for the memory bottleneck supporting speculative decoding at scale. However, Figure 1a appears to represent the combined "KV load and store" time rather than isolating the KV load time alone. Could you clarify what portion of this time is allocated to "KV store"? Understanding the distinction between load and store times would help validate the impact of the bottleneck more precisely.

### Comment on Lines 203-211 ("Expected Generation Length Per Step")
The preprint by Timor, Nadav, et al., titled *"Distributed Speculative Inference of Large Language Models"* (arXiv preprint arXiv:2405.14105, 2024), appears to be highly relevant to your study, particularly in examining the regime where SD leads to either speedups or slowdowns based on the drafting latency budget. Consider discussing this work to strengthen the related literature section.

---

> ### Author Response · Authors · 2024-11-21
> **Response to Reviewer UdZh (Part 1/2)**
>
> Thank you for providing such thoughtful and supportive feedback. We are glad that you found our analysis novel and insightful, and our contribution highly valuable. Based on your excellent suggestion, we have streamlined the analysis section discussing speculation decoding speedup factors and KV caching. We hope that we have been able to improve the readability of our paper, especially for the broader research community.
>
> Additionally, we are thankful for your insightful questions and further suggestions to improve our paper. We hope that responses answer some of your questions, and would look forward to any further comments.
>
> ---
>
> ### **Q1: Can you clarify the potential trade-offs in performance if MagicDec were applied to significantly smaller LLM models?**
>
> Interestingly, we find that smaller LLMs have a sharper growth in speedup with batch size compared to larger LLMs. In addition, the critical sequence length beyond which they attain higher speedups is also lower compared to the latter. For instance, while the critical sequence length for Llama-3.1-8B is ~4000, that of Llama-3.2-1B is just ~2000. This is because of two reasons:
> - The verification-to-decoding cost ratio is lower for smaller LLMs. For a given batch size and hardware, smaller LLMs become memory-bound for a shorter sequence length because of a smaller hidden state dimension. The more memory-bound the target model is, the smaller is the verification-to-decoding cost ratio.
> - The draft-to-target cost ratio is lower as well. Because the parameter and compute cost shared by the draft and the target model are much smaller than the cost the draft model optimizes for, the KV loading cost.
>
> Here is a comparison between the theoretical speedups achieved by Llama-3.2-1B model and Llama-3.1-8B model using self-speculation.
>
> **Speedups achieved by Llama-3.2-1B model**
>
> | prefill | bsz | gamma | accept rate | target_time | speedup |
> |---------|-----|-------|-------------|-------------|---------|
> | 16000   | 4   | 2     | 0.833       | 0.368218    | 1.18883 |
> | 16000   | 16  | 3     | 0.833       | 0.936144    | 1.58452 |
> | 16000   | 64  | 4     | 0.833       | 3.20145     | 1.98784 |
> | 16000   | 256 | 4     | 0.833       | 12.2499     | 2.15768 |
>
> **Speedups achieved by Llama-3.1-8B model**
>
> | prefill | bsz | gamma | accept rate | target_time | speedup |
> |---------|-----|-------|-------------|-------------|---------|
> | 16000   | 4   | 1     | 0.82        | 0.983983    | 1.08902 |
> | 16000   | 16  | 3     | 0.82        | 2.09072     | 1.39469 |
> | 16000   | 64  | 4     | 0.82        | 6.52403     | 1.84771 |
> | 16000   | 256 | 4     | 0.82        | 24.26       | 2.10895 |
>
> However, there could be one potential disadvantage for smaller LLMs. Even though self-speculation is usually sufficient for them, in general, they do not have suitable small draft model options. Hence, they might lose a little bit of flexibility in terms of choosing the best drafting strategy.
>
> ---
>
> ### **Q2: Is the performance gain from MagicDec sustained across varying types of long-context tasks, particularly those that require variable batch sizes or non-standard hardware configurations?**
>
> Since MagicDec presents an algorithm that can be deployed in different task and hardware settings, we believe that performance gains can be sustained across a broad application space. Moreover, MagicDec is flexible enough to adapt to varying types of task loads. For instance: If some long-context task requires variable batch sizes, then MagicDec would dynamically choose the appropriate draft KV budget and KV compression algorithm to achieve the best performance for the corresponding batch size. In case of non-standard hardware, MagicDec’s effectiveness depends on the peak FLOPs-to-memory bandwidth ratio of the device. For instance, for CPU-based inference, our method would suggest using regular autoregressive decoding instead.
>
> Please let us know if this answers your question. We would be happy to discuss this in more detail.

---

> ### Author Response · Authors · 2024-11-21
> **Response to Reviewer UdZh (Part 2/2)**
>
> ### **Comments**
>
> #### **Comment 1: Comment on Lines 52-74 and Figure 1a**
> Thanks for your suggestion. The KV store time is a very small portion of the total inference time. We have updated Figure 1a $\text{\textcolor{blue}{(Section 1, Page 2)}}$, isolating KV load time and store time to further clarify our claim. For a sequence with prefix length equal to 16000, during decoding the KV load time is approximately 16000 times larger than the store time, as each time we need to load all the previous tokens’ KV cache, while only need to store the key and value states of the new generated token. Thus, the bottleneck of inference is exactly KV load time.
>
> #### **Comment 2: Comment on Lines 203-211 ("Expected Generation Length Per Step")**
> Thanks for mentioning this interesting work. You are right this work also uses expected generation length and draft cost to assess whether a draft model is good or not. We think the distributed speculative inference proposed in this paper is perfectly complementary with our work. The distributed speculative inference overlaps verification cost. With the high acceptance rate and low draft cost offered by compressed KV-based drafting, the speedup could be higher when applied to long-context serving. We have added the discussion of this work in our related work section $\text{\textcolor{blue}{(Section 2, Page 3)}}$.

---

> ### Author Response · Authors · 2024-11-25
>
> Dear Reviewer UdZh,
>
> Thank you for your thoughtful and constructive feedback on our work. We have carefully addressed your comments and revised the work and manuscript accordingly. As the discussion period nears its end, we would greatly appreciate any additional questions or points of clarification. If our responses have satisfactorily addressed your concerns, we kindly ask you to consider reflecting this in your score.
>
> Thanks again for your time and expertise.

---

> ### Author Response · Authors · 2024-12-01
>
> Dear Reviewer UdZh,
>
> Thank you once again for your thoughtful feedback and the time you’ve dedicated to reviewing our work. As the extended discussion period draws to a close, we want to ensure that all your concerns have been fully addressed. If there are any remaining points requiring further clarification, please don’t hesitate to let us know.
>
> We deeply appreciate your time and valuable input, which have been instrumental in improving our work.

---

### Official Review · Reviewer_PRVw · 2024-11-04

**Soundness:** 3
**Presentation:** 2
**Contribution:** 2
**Rating:** 6
**Confidence:** 3

**Summary:**

The paper discusses the batched speculative decoding in the long context scenario. It finds a critical sequence length that is the threshold for batched speculative above which it can show speedups compared to the original autoregressive decoding. The paper further examines the compressed kv cache and tries to find the best compressed KV-based drafting with the best strategy.

**Strengths:**

- Interesting insights on the significant sequence length and how the nature of memory bound translates to speedups in batched speculative decoding setting.

- Insights that changing the KV drafting instead of the smaller draft models can potentially have better acceptance rate and thus better speedups.

**Weaknesses:**

- Perhaps on the incremental side since the main new ideas are not all that large. The idea of compression KV is already in the literature.

- Perhaps the paper jump to some conclusions too fast without enough explanations, which makes it a little hard to follow.

  - What does Figure 1(b) try to show?

  - Maybe consider adding more details about the effects of increasing batch size with the original speculative decoding.


- Some terms are vague in the texts. Authors may consider more clearly define them.

   - What does Figure 1(b) try to show?

  - What does ``self-speculation" in the texts mean?

  -  What does KV budget mean?

- Some important design details can be more explicitly clarified.

  - Where does the time breakdown such as Figure 1(a) come from? GPU kernels can hide latency among threads so the end-to-end time does not necessarily equals to the combination of the memory loading and computing time. Nevertheless, the intuition of memory bound of KV cache when batch size is large makes sense.

  - Is KV compression used for both the draft and target model?

  - What is the baseline for results in Figure 4?

  - As for this lossy KV compression selection method, perhaps consider adding the evaluation that shows the selection in KV compression strategy has advantage (e.g., speedups) over the original speculative decoding.

  - How the framework guide the selection of KV compression method remains unclear to me.

- The applications for the very long contexts prompts in large batch size still need further justification.

**Questions:**

See above.

---

> ### Author Response · Authors · 2024-11-21
> **Response to Reviewer PRVw (Part 1/4)**
>
> We thank the reviewer's suggestions and questions. We have updated the manuscript to clarify the novelty of MagicDec **[Q1]**, add more explanation for Figure 1(b) and details of increasing batch size with original speculative decoding **[Q2]**. We also added more explanations for some terms in the paper **[Q3]** and clarified the details mentioned by the reviewer **[Q4]**. We add some experiments on smaller batch sizes and shorter context length settings to show the effectiveness of MagicDec for not very long prompts **[Q5]**. We hope our detailed clarification with further experiment results will clear the doubts about the significance of our work.
>
> ---
>
> ### **Q1: The idea of compression KV is already in the literature, so the main new ideas are not all that large.**
>
> Thanks for pointing out the lack of clarity about the primary novelty of our work. We want to first clarify the main contributions of MagicDec. The conventional wisdom says that speculative decoding can not provide speedup for large batch inference, primarily because the token verification process becomes too expensive in the compute-bound regime. We are the first to present the limitations of this existing wisdom.
>
> - Through the analysis of speculative decoding speedup and LLM inference performance, **MagicDec first identifies for long-context serving, speculative decoding can accelerate large batch inference. More interestingly, the speedup even increases with batch size**.
> - **MagicDec proposes the key to achieve high speedup is keeping the draft cost growing independently with sequence length.** KV cache is the performance bottleneck that scales with both batch size and context length, so compressing the KV cache of the draft model can be a good way to limit the draft cost. Thus identifying KV compression as a necessary tool for efficient drafting is our main novelty rather than proposing a new compression method. There are potentially several ways to achieve that including small draft models with compressed KV cache, original model speculating itself with compressed KV cache or skipping its own layers (as suggested by R1) etc.
> - Finally, the primary goal of KV compression has traditionally been to preserve model accuracy. However, it remains unclear whether higher model accuracy directly correlates with higher token acceptance rates. For instance, while Llama-3.1-70B is more accurate than Llama-3.1-8B, it exhibits a lower token acceptance rate when speculating the latter. Interestingly, MagicDec suggests some KV compression algorithms can indeed achieve token acceptance rates when used in drafting stages.
>
> Hence, all the existing KV compression techniques are indeed multiple ways to help us achieve the goal – keep draft cost constant with sequence length. **MagicDec is a general framework that guides how to choose the optimal drafting strategy or KV compression methods based on draft cost, acceptance rate and hardware**.

---

> ### Author Response · Authors · 2024-11-21
> **Response to Reviewer PRVw (Part 2/4)**
>
> ### **Q2: The paper jumps to some conclusions too fast without enough explanations**
>
> Thanks for pointing out the writing issue. We have revised our paper to make our explanations more streamlined and easy to follow. The modifications are shown in blue color in the **Introduction** section. Specifically:
>
> #### **What does Figure 1(b) try to show?**
> Figure 1(b) illustrates the comparison of throughput between MagicDec and standard autoregressive decoding at a given token-wise latency budget. It is well-known that simultaneously improving throughput and latency is challenging, especially when model quality cannot be sacrificed. This plot exhibits MagicDec’s ability to achieve better throughput and latency across the spectrum for long-context requests.
>
> #### **Maybe consider adding more details about the effects of increasing batch size with the original speculative decoding.**
> Thanks for figuring this out. We have added additional experiment results of original speculative decoding in the paper to show the effect of increasing batch size, which is shown in Figure 7(a) $\text{\textcolor{blue}{(Fig. 7, Page 10)}}$. The detailed results are shown below. **T_D** stands for draft cost.**T_V** stands for verification cost. **T_SD** is the average speculative decoding latency (ms), while **T_Auto** is the baseline autoregressive decoding latency. We use Llama-3.2-1B as the draft model to speculate Llama-3.1-8B. The results show that **speedup decreases as batch size increases.**
>
> | Bsz | Prompt Len | Gamma | T_D  | T_V   | Acc_Len | T_SD  | T_Auto | Speedup |
> |-----|------------|-------|------|-------|---------|-------|--------|---------|
> | 16  | 256        | 3     | 4.32 | 3.79  | 2.75    | 3.21  | 3.29   | 1.02    |
> | 32  | 256        | 2     | 3.18 | 4.01  | 2.34    | 3.36  | 3.59   | 1.07    |
> | 64  | 256        | 2     | 3.43 | 4.60  | 2.29    | 3.83  | 3.91   | 1.02    |
> | 128 | 256        | 1     | 2.03 | 5.03  | 1.67    | 4.68  | 4.35   | 0.93    |
> | 256 | 256        | 1     | 2.45 | 6.91  | 1.72    | 5.91  | 5.23   | 0.88    |
> | 512 | 256        | 1     | 3.40 | 10.57 | 1.74    | 8.62  | 7.36   | 0.85    |
>
> ---
>
> ### **Q3: Some terms are vague in the texts.**
>
> We thank the reviewer for figuring out these issues. We have updated more explanation for the terms mentioned by the reviewer in the paper (highlighted blue). Specifically:
>
> #### **What does “self-speculation” in the texts mean?**
> Self-speculation in our work refers to leveraging the same LLM to perform speculative decoding by utilizing a compressed KV cache as a draft mechanism. During the drafting stage, we use the LLM with the compressed KV cache to generate several tokens. During the verification stage, we use the LLM with the full KV cache to verify these drafted tokens.
>
> #### **What does “KV budget” mean?**
> KV budget in our paper means the size of the KV cache for each sequence after compression.

---

> ### Author Response · Authors · 2024-11-21
> **Response to Reviewer PRVw (Part 3/4)**
>
> ### **Q4. Some important design details can be more explicitly clarified.**
>
> We thank the reviewer for posing these questions. We have added clarifications for the details mentioned by the reviewer with blue color in the revised paper. Specifically:
>
> #### **Where does the time breakdown such as Figure 1(a) come from?**
> The time breakdown is based on the analysis results from [1], which is a tool for visualizing LLMs and analyzing the performance on different hardware platforms. We have used both roofline modeling (as suggested by you) and additive modeling (does not consider latency-hiding optimizations) in our analysis. However, in our paper, we report the time breakdown with additive modeling only, as neither of the two theoretical modeling approaches is fully accurate. As you have mentioned, the modeling type does not contradict the main takeaways of the paper.
> **[1]** Yuan Z, Shang Y, Zhou Y, et al. LLM Inference Unveiled: Survey and Roofline Model Insights. arXiv preprint arXiv:2402.16363, 2024.
>
>
> #### **Is KV compression used for both the draft and target model?**
> The KV compression is only used for the draft model, while the draft model can be a smaller model or the target model itself (self-speculation). For self-speculation, we actually have two independent KV caches, but use the same model weights. During the prefill phase of the target model, we use the KV compression method to generate a compressed KV cache with a certain KV budget. During the drafting stage, we use the LLM and the compressed KV cache to generate draft tokens. During the verification stage, we use the LLM with the full KV cache to verify the draft tokens.
>
> #### **What is the baseline for results in Figure 4?**
> In Figure 4, we don’t compare the results with any baseline. We can provide more explanations for Figure 4. Figure 4(a) and Figure 4(b) show that for different prompt lengths, the memory footprints of a small draft model will be close to or even surpass the target model when batch size increases. The increase of the draft KV cache leads to this phenomenon, demonstrating the inefficiency of a small draft model in large batch size inference regimes and the necessity of KV compression for the draft model. Figure 4(c) compares the acceptance rates of different KV compression methods when applied as the drafter in speculative decoding. Top-K in this context represents the theoretical upper bound of sparse attention approximation methods.
>
> #### **As for this lossy KV compression selection method, perhaps consider adding the evaluation that shows the selection in KV compression strategy has advantage (e.g., speedups) over the original speculative decoding.**
> We have added the comparison between the KV compression-based speculative decoding and the original speculative decoding to Figure 7(b) $\text{\textcolor{blue}{(Page 10, Evaluation Section)}}$ in the revised paper. The results are shown below. The input prompt length is 8192. Hardware: 8xH100. We use Llama-3.2-1B as the drafter to do speculation for Llama-3.1-8B.
>
> **Llama-3.2-1B, full KV cache**
> | Bsz | Gamma | T_D  | T_V   | Acc_Len | T_SD  | T_Auto | Speedup |
> |-----|-------|------|-------|---------|-------|--------|---------|
> | 16  | 2     | 3.52 | 4.64  | 2.33    | 3.79  | 4.12   | 1.09    |
> | 64  | 2     | 4.92 | 7.33  | 2.23    | 5.82  | 6.62   | 1.14    |
> | 128 | 2     | 6.62 | 11.47 | 2.24    | 8.42  | 9.68   | 1.15    |
> | 256 | 2     | 9.78 | 19.51 | 2.24    | 13.45 | 15.86  | 1.18    |
> | 480 | OOM   | OOM  | OOM   | OOM     | OOM   | OOM    | OOM     |
>
> **Llama-3.2-1B, compressed KV with Constant Budget 512**
> | Bsz | Gamma | T_D  | T_V   | Acc_Len | T_SD  | T_Auto | Speedup |
> |-----|-------|------|-------|---------|-------|--------|---------|
> | 16  | 2     | 3.02 | 4.63  | 2.25    | 3.70  | 4.12   | 1.11    |
> | 64  | 2     | 3.45 | 7.33  | 2.14    | 5.38  | 6.62   | 1.23    |
> | 128 | 2     | 3.97 | 11.47 | 2.16    | 7.49  | 9.68   | 1.29    |
> | 256 | 3     | 6.65 | 20.73 | 2.48    | 11.40 | 15.86  | 1.39    |
> | 480 | 3     | 8.79 | 38.28 | 2.47    | 19.51 | 27.53  | 1.41    |
>
> We can see from these results that Llama-3.2-1B with compressed KV cache outperforms original speculative decoding for each batch size. The main reason for this is that compressed KV cache limits the growth of draft cost while still keeping a high acceptance rate.
>
> #### **How the framework guides the selection of KV compression method remains unclear to me.**
> Thanks for pointing out the unclarity. We have updated the Method section $\text{\textcolor{blue}{(Section 4, Pages 7-8)}}$ of the revised paper to better illustrate how to choose the optimal drafting strategy from: Different draft model sizes, Draft KV budgets, KV compression methods.

---

> ### Author Response · Authors · 2024-11-21
> **Response to Reviewer PRVw (Part 4/4)**
>
> ### **Q5. The applications for the very long contexts prompts in large batch size still need further justification.**
>
> We thank the reviewer for raising the concern of real-world application for very long-context and large batch size prompts. As we have mentioned in our response to **Reviwer 3wSP**, the emergence of long-context models, retrieval-augmented generation (RAG), and in-context learning has dramatically increased prompt lengths in recent years. (**Anthropic [1]**). Even if the user prompts are not too long, the production systems internally increase the context length with retrieved documents or chain-of-thought prompting. For instance, recently OpenAI o1 has started using COT prompting to improve their responses.
>
> But, we agree that prompt length like 100k with large batch size may not be as common in current real-world applications. However, MagicDec can also get gain for moderate length input prompt and medium batch size without adding any cost or hurting accuracy. As shown in Figure 3 (Page 5) of our paper, for 8xA100 GPU with a batch size of 256, speculative decoding becomes beneficial once the context length exceeds 3297 tokens, a threshold we term the **critical sequence length**. For modern GPUs in cloud centers like the H100, this critical length is even lower due to the higher FLOPs-to-memory bandwidth ratio. This critical prompt length is not a very large value, and is more common in real-world applications.
>
> To demonstrate practical applicability, we tested MagicDec using Llama-3.2-1B with streamingLLM KV cache for speculation of Llama-3.1-8B. The draft budget is 256. We used a prompt length of 3072 and a batch size of 128. Results are shown below:
>
> | GPU    | Bsz | Prompt Len | Gamma | T_drf | T_ver | acc_len | T_sd  | T_auto | Speedup |
> |--------|-----|------------|-------|-------|-------|---------|-------|--------|---------|
> | 8xH100 | 128 | 3072       | 1     | 2.03  | 6.97  | 1.70    | 5.73  | 6.27   | 1.09    |
>
> For this moderate sequence length and batch size, MagicDec demonstrates speedup over standard decoding without hurting generation quality. And as batch size or sequence length increases, the speedup becomes even more pronounced.
>
> **[1]** Anthropic: Prompt Caching with Claude. https://www.anthropic.com/news/prompt-caching

---

> ### Author Response · Authors · 2024-11-25
>
> Dear Reviewer PRVw,
>
> Thank you for your thoughtful and constructive feedback on our work. We have carefully addressed your comments and revised the work and manuscript accordingly. As the discussion period nears its end, we would greatly appreciate any additional questions or points of clarification. If our responses have satisfactorily addressed your concerns, we kindly ask you to consider reflecting this in your score.
>
> Thanks again for your time and expertise.

---

> ### Author Response · Authors · 2024-12-01
>
> Dear Reviewer PRVw,
>
> Thank you once again for your thoughtful feedback and the time you’ve dedicated to reviewing our work. As the extended discussion period draws to a close, we want to ensure that all your concerns have been fully addressed. If there are any remaining points requiring further clarification, please don’t hesitate to let us know.
>
> We deeply appreciate your time and valuable input, which have been instrumental in improving our work.

---

> > ### Comment · Reviewer_PRVw · 2024-12-02
> >
> > Dear Authors,
> >
> > Thanks for clarifying the questions regarding novelty and design details. The answers are carefully written and are in my opinion convincing.

---

### Official Review · Reviewer_XfUb · 2024-11-04

**Soundness:** 2
**Presentation:** 4
**Contribution:** 3
**Rating:** 5
**Confidence:** 3

**Summary:**

This paper introduces a speculative decoding technique designed to enhance throughput and reduce latency in long-context Large Language Models (LLMs), addressing a common bottleneck in high-batch, memory-bound inference tasks. By optimizing the Key-Value (KV) cache through sparse KV configurations and employing self-speculation, the method improves memory efficiency, making speculative decoding effective for large batch sizes and moderate-to-long sequence lengths. Empirical results on high-performance GPUs, such as A100s and H100s, show up to a 2.51x speedup compared to traditional autoregressive decoding, especially in scenarios with long sequences benchmarking. While promising, the approach lacks comparisons of some baselines, evaluating a limited spectrum of model sets, lack of case study or worse case analysis, and lacks a discussion of the scope/limitation section. If the authors could address the concerns in revision, I would be willing to raise the score.

**Strengths:**

1. The writing is pretty good and easy to follow.

2. Sufficient experiments conducted on high-performance GPUs (e.g., A100, H100) show up to a 2x speedup compared to autoregressive decoding, demonstrating speculative decoding’s efficiency for long sequences.

3. The authors provide a detailed mathematical analysis showing how speculative decoding can be effective even for large batch sizes in memory-bound regimes, particularly by addressing the KV cache bottleneck.

**Weaknesses:**

1. Missing discussion and comparisons of reasonable baselines. Although the authors briefly discuss TriForce, which demonstrates the effectiveness of self-speculation with compressed KV, they did not compare it with TriForce in experimentation for some reasons. Besides, there exists some self-speculation work aiming to accelerate LLM inference, e.g.,  Xia et al. [1] and Zhang et al. [2]. Could such existing works be applicable to / integrated with the proposed solution? Please either provide satisfactory reasons why forgiving such baselines or add comparisons in experiments and clear this work's novelty. While the paper references approaches like vLLM, it lacks a side-by-side comparison, especially regarding memory efficiency versus other batch-processing strategies. Including metrics from existing solutions in similar conditions would solidify the paper’s contributions.

[1] SWIFT: ON-THE-FLY SELF-SPECULATIVE DECODING FOR LLM INFERENCE ACCELERATION

[2] Draft & Verify: Lossless Large Language Model Acceleration via Self-Speculative Decoding

2. Evaluation of a limited spectrum of models. The experimental settings focus on LLaMA series models, yet the performance of proposed methods on other models remains unknown.

3. Lack of Case study or worse case analysis. The paper describes handling variability in token acceptance rates during speculation, but it could provide more details on failure cases or worst-case scenarios where rejection rates could impair throughput significantly. This is especially important for practical deployments in heterogeneous batches where sequence lengths vary widely.

3. Lack of discussion and limitation. Providing a brief discussion of limitations would help clear the scope of this work and make broader impacts. For instance, this work focuses on high-end modern GPUs, so the proposed solution may not perform that well on desktop/low-end GPUs, or even worse at resource-constrained embedded systems. Including tests on more widely accessible hardware (e.g., T4 or V100 GPUs) would demonstrate the method’s practical viability across a broader range of settings. The authors could elaborate more directions in the revisions.

**Questions:**

See weakness part.

---

> ### Author Response · Authors · 2024-11-21
> **Response to Reviewer XfUb (Part 1/4)**
>
> We appreciate your supportive comments and constructive suggestions. We have updated our manuscript to clarify the confusion about “missing baselines” **[Q1]** and have added comprehensive evaluations across different models **[Q2]** to illustrate the generalizability of our approach. In addition, we have added more case studies **[Q3]** regarding “worst case analysis” and stated the limitations of our current setup **[Q4]**. We hope our detailed clarification with further experiment results will clear the doubts about the significance of our work.
>
> ---
>
> ### **Q1. Missing discussion and comparisons of reasonable baselines: Triforce and other self-speculation methods, inference pipelines like VLLM**
>
> We thank the reviewer for suggesting these self-speculation works. We first want to clarify that our work illustrates how speculative decoding can be made useful even in a large batch setting and provides a generalized framework to evaluate different speculative decoding algorithms in a long-context large batch size regime.  **MagicDec requires a drafting strategy whose KV cache loading cost increases slower than the target model with increasing batch size and sequence length.** Hence, any KV compression method for draft KV cache is an ideal candidate for MagicDec.
>
> From that perspective, the KV retrieval algorithm of Triforce does indeed fall under the suite of algorithms that MagicDec can work with. Hence, it actually complements our efforts rather than serving as a baseline.
>
> We also appreciate the other useful self-speculation methods suggested, such as Xia et al. [1] and Zhang et al. [2], which utilize layer-skipping strategies. These methods can also reduce draft KV cache cost and hence can be added on top of the retrieval algorithms that MagicDec studies. In summary, we view these suggested methods as complementary efforts that can expand MagicDec’s search space of suitable KV compression algorithms rather than as baselines.
>
> Finally, MagicDec can provide additional performance improvements when deployed on efficient inference frameworks like VLLM by optimizing the inference-time memory loading cost. In fact, continuous batching-based serving systems allow MagicDec to perform at its full potential. Because continuous batching techniques have high memory utilization, they represent the exact setting where MagicDec can be most effective. Although our current implementation does not support continuous batching, we are planning to integrate MagicDec into VLLM in our future work.
>
> **[1]** SWIFT: On-the-Fly Self-Speculative Decoding for LLM Inference Acceleration (Xia et al. 2024)
>
> **[2]** Draft & Verify: Lossless Large Language Model Acceleration via Self-Speculative Decoding (Zhang et al.)

---

> ### Author Response · Authors · 2024-11-21
> **Response to Reviewer XfUb (Part 2/4)**
>
> ### **Q2. Evaluation on Limited Spectrum of Models**
> Thanks for suggesting us to evaluate MagicDec on a broader spectrum of models. We have added evaluations for Mistral and Qwen series models to show the trends seen for Llama models also translate to the former $\text{\textcolor{blue}{(Appendix A.5, Page 15)}}$.
>
> #### **MagicDec achieves impressive speedups for Mistral-7B-v0.3, Qwen-2.5-7B and Qwen2.5-32B even at large batch sizes**
>
> We utilize self-speculation with SnapKV based KV selection for Mistral-7B-v0.3 and Qwen-2.5-7B models. For Qwen2.5-32B, we utilize Qwen-2.5-7B with streamingLLM KV cache as the draft model. We report the speedups obtained with the optimal speculation lengths.
>
> > Note: we use 8xH100s for Mistral-7B-v0.3 and Qwen-2.5-32B models and 4xH100s for Qwen-2.5-7B because it has 4 KV heads.
>
> | Model           | Bsz | Gamma | T_draft | T_verification | mean_accepted_length | T_specdec | T_autoregressive | Speedup |
> |------------------|-----|-------|---------|----------------|-----------------------|-----------|------------------|---------|
> | Mistral-7B-v0.3 | 128 | 5     | 27.49   | 30.65          | 4.72                  | 12.31     | 25.41            | 2.06    |
> | Qwen-2.5-7B     | 128 | 5     | 27.22   | 28.51          | 4.62                  | 12.06     | 22.79            | 1.89    |
> | Qwen-2.5-32B    | 32  | 3     | 11.78   | 21.82          | 2.62                  | 12.85     | 19.43            | 1.51    |
>
> #### **Similar to Llama models, we can see **increasing speedups with increasing batch size** for Mistral and Qwen models**
>
> Mistral-7B-v0.3 Speedups with SnapKV-Based Self-Speculation (Using 8xH100s)
>
> | Bsz | Gamma | T_draft | T_verification | mean_accepted_length | T_specdec | T_autoregressive | Speedup |
> |-----|-------|-------|-------|---------|------|--------|---------|
> | 32  | 3     | 11.71 | 9.62  | 3.49    | 6.12 | 8.92   | 1.46    |
> | 64  | 3     | 13.64 | 15.64 | 3.47    | 8.44 | 14.49  | 1.72    |
> | 128 | 5     | 27.49 | 30.65 | 4.72    | 12.31| 25.41  | 2.06    |
>
> Qwen-2.5-7B Speedups with SnapKV-Based Self-Speculation (Using 4xH100s)
>
> | Bsz | Gamma | T_draft | T_verification | mean_accepted_length | T_specdec | T_autoregressive | Speedup |
> |-----|-------|-------|-------|---------|------|--------|---------|
> | 32  | 3     | 11.40 | 9.26  | 3.40    | 6.07 | 8.20   | 1.35    |
> | 64  | 4     | 17.67 | 15.67 | 4.06    | 8.20 | 13.11  | 1.60    |
> | 128 | 5     | 27.22 | 28.51 | 4.62    | 12.06| 22.79  | 1.89    |
>
> Qwen-2.5-7B Speedups with Qwen-2.5-7B Draft Using StreamingLLM Cache (Using 4xH100s)
>
> | Bsz | Gamma | T_draft | T_verification | mean_accepted_length | T_specdec | T_autoregressive | Speedup |
> |-----|-------|-------|-------|---------|------|--------|---------|
> | 8   | 2     | 6.77  | 11.31 | 2.27    | 7.97 | 10.42  | 1.31    |
> | 16  | 2     | 7.21  | 14.59 | 2.26    | 9.64 | 13.36  | 1.39    |
> | 32  | 3     | 11.78 | 21.82 | 2.62    | 12.85| 19.43  | 1.51    |

---

> ### Author Response · Authors · 2024-11-21
> **Response to Reviewer XfUb (Part 3/4)**
>
> ### **Q3. Lack of Case Study or Worse-Case Analysis**
> We appreciate the reviewer’s concern about the variability in token acceptance rate as it is an important consideration in our speedup analysis. Our analysis shows that for a fixed setting, i.e. model, hardware, draft cost etc., different sequence length requires different minimum acceptance rates to achieve any speedup with self-speculation, with longer context lengths having more relaxed requirements.
>
> The following table illustrates for a given draft KV cache budget, the minimum acceptance rate required by different sequence lengths to see any speedup (we consider batch size 4 here) for self-speculation. The admissibility of the draft budgets for each setting is based on the empirical acceptance rates obtained for PG-19 documents. The min_acceptance_rate for the small draft model with compressed KV cache will be much lower than these values because of lower draft_cost.
>
> |   prefill length |   draft KV cache size |   min_acceptance_rate | admissible   |
> |------------------:|----------------------:|-----------------------:|:-------------|
> |              8000 |                  128 |                  0.916 | No           |
> |              8000 |                  256 |                  0.918 | No           |
> |              8000 |                  512 |                  0.921 | No           |
> |              8000 |                 1024 |                  0.928 | No           |
> |             12000 |                  128 |                  0.878 | No           |
> |             12000 |                  256 |                  0.880 | No           |
> |             12000 |                  512 |                  0.883 | **Yes**          |
> |             12000 |                 1024 |                  0.890 | **Yes**          |
> |             16000 |                  128 |                  0.846 | No           |
> |             16000 |                  256 |                  0.848 | **Yes**          |
> |             16000 |                  512 |                  0.851 | **Yes**          |
> |             16000 |                 1024 |                  0.857 | **Yes**          |
> |             20000 |                  128 |                  0.817 | No           |
> |             20000 |                  256 |                  0.818 | **Yes**          |
> |             20000 |                  512 |                  0.821 | **Yes**          |
> |             20000 |                 1024 |                  0.828 | **Yes**          |
>
> This indicates that a single KV cache budget can not be applicable for all sequence lengths.
>
> #### **Challenges for Heterogeneous Batches and Ways to Mitigate Them**
> As you have pointed out, the phenomenon discussed above is an important consideration in a real-world setting where heterogeneous batches can appear with sequences of varying lengths. Our analysis above can guide us to overcome this challenge in the following ways:
> - **Allocating Different Draft KV Cache Budgets for Sequences of Different Lengths:**
>   Because MagicDec supports PagedAttention, different draft KV cache budgets can be easily allocated to different requests in the batch.
> - **Request-Scheduling Algorithms:**
>   We can develop intelligent request scheduling algorithms based on their sequence lengths and task type (often dictates the acceptance rate). Recent works like Fu et al. [1], Srivatsa et al. [2] have developed new algorithms to schedule requests across multiple computing instances for better load balancing and higher throughput. Similarly, we can support different drafting algorithms with different draft cost budgets on different computing nodes and route the incoming requests to suitable nodes based on their requirements.
> - **Adding More Flexibility to Real-World Distributed Serving Systems:**
>   Recent distributed serving systems (Lin et al. [3]) have looked into distributing attention computation by offloading KV caches to a shared GPU pool in the same computing cluster. This approach has been effective for better load balancing and improving throughput of real-world heterogeneous batches, although at the cost of some communication overhead of routing query vectors to perform attention. In this scenario, MagicDec can provide new opportunities by prioritizing retention of draft KV cache in local memory and only utilizing the offloaded KV cache for verification purposes.
>
> #### **References**
> [1] Efficient LLM Scheduling by Learning to Rank (Fu et al., 2024)
> [2] Preble: Efficient Distributed Prompt Scheduling for LLM Serving (Srivatsa et al., 2024)
> [3] Infinite-LLM: Efficient LLM Service for Long Context with DistAttention and Distributed KVCache (Lin et al., 2024)

---

> ### Author Response · Authors · 2024-11-21
> **Response to Reviewer XfUb (Part 4/4)**
>
> ### **Q4. Lack of Discussion and Limitation: Performance Analysis on Lower-End GPUs**
> Thanks for the suggestion. In our revised draft, we have added the limitations of our work $\text{\textcolor{blue}{(Section 6, Page 10)}}$, that can be summarized as follows:
> - Higher-end GPUs benefit more from MagicDec because of their better peak FLOPs to memory bandwidth ratio.
> - Our current work only focuses on improving decoding performance.
> - Our work does not exhaustively study all the SOTA KV compression algorithms, restricting ourselves to fewer algorithms that are representative of a broader class of KV compression methods.
>
> #### **Performance on Commodity Machines**
> MagicDec works better with GPUs with high peak FLOPs to memory bandwidth ratio. Because LLM inference is more memory-bound on such devices, restricting verification to target decoding cost ratio to a value close to 1. We have compared MagicDec’s speedups on customer-level machine 4090 with that on H100. For these experiments, we used Llama-3.2-1B with StreamingLLM KV as the draft model to speculate Llama-3.1-8B.
>
> | GPU    | Batch Size | Gamma | T_draft | T_verification | mean_accepted_length | T_specdec | T_autoregressive | Speedup |
> |--------|------------|-------|---------|----------------|-----------------------|-----------|------------------|---------|
> | 8xH100 | 16         | 3     | 4.43    | 6.71           | 2.43                  | 4.86      | 6.18             | 1.27    |
> | 8x4090 | 16         | 2     | 5.95    | 20.65          | 2.10                  | 13.18     | 15.92            | 1.21    |
> | 8xH100 | 32         | 3     | 4.71    | 9.70           | 2.43                  | 6.22      | 9.10             | 1.46    |
> | 8x4090 | 32         | 2     | 7.52    | 33.90          | 2.13                  | 19.99     | 27.40            | 1.37    |
>
> If the inference system utilizes an even resource-constrained device with a much smaller peak FLOPs to memory bandwidth ratio, MagicDec would suggest to use autoregressive decoding instead if required.
>
> **A Possible Utilization of MagicDec on Commodity Machines**
> We present a way to adapt MagicDec to serve LLMs using commodity machines. To support batch inference of long sequences on these lower-end devices, we can adopt a distributed attention strategy proposed by recent works like [5, 6, 7] in order to optimize resource utilization. These approaches involve offloading part of the KV cache to other devices, including CPU. As described before in Q3, MagicDec can provide new opportunities to reduce the communication overhead in these methods by locally accommodating the compressed KV cache for drafting and only utilizing distributed attention for verification purposes. The nature of speculative decoding can also decrease the run times of the large target model with full KV cache, thus amortizing the communication overhead in offloading or distributed settings.
>
> #### **Optimizing Decoding Performance Only**
> Our current work only focuses on decoding performance, without any concern for prefilling performance. However, considering a disaggregated serving system (Zhong et al. [1], Qin et al. [2]) that decouples prefill and decoding stages, optimizing decoding performance alone is also quite beneficial.
>
> #### **Non-Exhaustive Study of KV Compression Methods**
> Our method is not an exhaustive collection of all the state-of-the-art KV compression techniques. Instead, we pick some representative solutions (static and dynamic KV selection strategies) based on their empirical acceptance rates to demonstrate the trade-off analysis among different kinds of KV compression methods in different batch size and sequence length regimes.
>
> #### **Future Plan**
> We plan to conduct an extensive study across all the state-of-the-art SD variants, including the layer-skip methods like (Elhoushi et al. [3], Zhang et al. [4], Xia et al. [5]), KV quantization (Liu et al. [6], Hooper et al. [7]), low rank decomposition (Singhania et al. [8]), etc., that are likely to mitigate the KV bottleneck problem in a large batch size, long-context regime.
>
> #### **References**
> **[1]** DistServe: Disaggregating Prefill and Decoding for Goodput-optimized Large Language Model Serving (Zhong et al., 2024)
> **[2]** Mooncake: A KVCache-centric Disaggregated Architecture for LLM Serving (Qin et al., 2024)
> **[3]** LayerSkip: Enabling Early Exit Inference and Self-Speculative Decoding (Elhoushi et al., 2024)
> **[4]** Draft & Verify: Lossless Large Language Model Acceleration via Self-Speculative Decoding (Zhang et al., 2023)
> **[5]** SWIFT: On-the-Fly Self-Speculative Decoding for LLM Inference Acceleration (Xia et al., 2024)
> **[6]** KIVI: A Tuning-Free Asymmetric 2bit Quantization for KV Cache (Liu et al., 2024)
> **[7]** KVQuant: Towards 10 Million Context Length LLM Inference with KV Cache Quantization (Hooper et al., 2024)
> **[8]** Loki: Low-rank Keys for Efficient Sparse Attention(Singhania et al., 2024)

---

> ### Author Response · Authors · 2024-11-25
>
> Dear Reviewer XfUb,
>
> Thank you for your thoughtful and constructive feedback on our work. We have carefully addressed your comments and revised the work and manuscript accordingly. As the discussion period nears its end, we would greatly appreciate any additional questions or points of clarification. If our responses have satisfactorily addressed your concerns, we kindly ask you to consider reflecting this in your score.
>
> Thanks again for your time and expertise.

---

> > ### Author Response · Authors · 2024-12-01
> >
> > Dear Reviewer XfUb,
> >
> > Thank you once again for your thoughtful feedback and the time you’ve dedicated to reviewing our work. As the extended discussion period draws to a close, we want to ensure that all your concerns have been fully addressed. If there are any remaining points requiring further clarification, please don’t hesitate to let us know.
> >
> > We deeply appreciate your time and valuable input, which have been instrumental in improving our work.

---

> ### Comment · Reviewer_XfUb · 2024-12-02
>
> Dear Authors,
>
> Thanks for clarifying the questions to address my concerns, and I decided to keep my score.

---

> > ### Author Response · Authors · 2024-12-03
> >
> > Dear Reviewer XfUb,
> >
> > Thank you for your feedback and for engaging with our responses. We are glad that our answers addressed your concerns. However, we noticed a shift in your rating from 6 to 5. Could you kindly share the reason behind this change? If there are any unresolved concerns, we would be happy to clarify further.
> >
> > We greatly appreciate your time and valuable insights.

---

> > > ### Comment · Reviewer_XfUb · 2024-12-03
> > >
> > > There are still some unresolved concerns.
> > >
> > > 1. Lack of Novelty: I remain concerned about the novelty of the proposed approach. It appears to be a combination of existing techniques, such as the KV cache compression and speculative decoding methods, which, when combined, seem relatively straightforward and lack significant innovation.
> > >  2. Experimental Validation: Another major concern is the lack of validation on larger or more comprehensive datasets. The absence of experiments on datasets such as infiniteBench and Ruler raises questions about the method’s generalizability and practical applicability. The incomplete experimental setup limits the robustness of the findings. Also, the authors lack comparisons with more baselines, which reflects their experimental settings are controlled and inpractical to realistic scenarios.
> > >
> > > Based on these two major concerns, I suggest that the authors incorporate these suggestions in future versions, and thanks for the time to address my previous concerns.

---

> ### Author Response · Authors · 2024-12-03
>
> Dear Reviewer XfUb,
>
> Thank you for sharing your concerns. We would like to further clarify them.
>
> ### **(1) Lack of Novelty**
>
> As we clarified in our response to Reviewer PRVw, the main contribution of MagicDec is its identification that speculative decoding can accelerate large-batch inference in long-context serving. This challenges the conventional wisdom that speculative decoding does not provide speedup for large-batch inference. **Reviewer UdZh, 3wSP and PRVw all acknowledged our novelty**.
>
> - Through the analysis of speculative decoding speedup and LLM inference performance, **MagicDec first identifies for long-context serving, speculative decoding can accelerate large batch inference.** More interestingly, the speedup even increases with batch size.
> - **MagicDec proposes the key to achieve high speedup is keeping the draft cost growing independently with sequence length.** KV cache is the performance bottleneck that scales with both batch size and context length, so compressing the KV cache of the draft model can be a good way to limit the draft cost. Thus identifying KV compression as a necessary tool for efficient drafting is our main novelty rather than proposing a new compression method. **There are potentially several ways to achieve that including small draft models with compressed KV cache, original model speculating itself with compressed KV cache or skipping its own layers etc.**
> - Finally, the primary goal of KV compression has traditionally been to preserve model accuracy. However, it remains unclear whether higher model accuracy directly correlates with higher token acceptance rates. For instance, while Llama-3.1-70B is more accurate than Llama-3.1-8B, it exhibits a lower token acceptance rate when speculating the latter. Interestingly, MagicDec suggests some KV compression algorithms can indeed achieve token acceptance rates when used in drafting stages.
>
> **Hence, all the existing KV compression techniques are indeed multiple ways to help us achieve the goal – keep draft cost constant with sequence length.** MagicDec is a general framework that guides how to choose the optimal drafting strategy or KV compression methods based on draft cost, acceptance rate and hardware.
>
> ### **(2) Experimental Validation**
>
> 1. We provided the experimental results on various tasks in our paper. We compare the acceptance rate of SnapKV-based self-speculation $\text{\textcolor{blue}{ (Figure 5, Section 4.1, Page 7)}}$ and the end-to-end speedup $\text{\textcolor{blue}{ (Table 2, Section 5.2, Page 9)}}$ comparison between PG-19, Ruler tasks(niah-multikey-3, cwe, qa-1). These results demonstrate the generalizability and practical applicability of MagicDec. In addition, we have not added results on InfiniteBench in particular, because most of the tasks in this benchmark have very small generated token lengths, which is not suitable to evaluate acceptance rates.
> > Update: We evaluated the acceptance rates of SnapKV based acceptance rates of Llama-3.1-8B on math-calc and longbook-sum-eng (the two subtasks of infinitebench with sufficiently long average output tokens). We got acceptance rates of 93.2% and 92.6% respectively for prefill length of 32k and draft budget 2k, **which are both higher than the acceptance rate on PG-19**.  **Based on our analysis, under the same prefill length, batch size and draft budget, the speedup of MagicDec on math-calc and longbook-sum-eng tasks of Infinitebench is expected to be higher than the speedup on PG-19 when using the same KV-compression method due to the higher acceptance rate.** For instance, we achieved a **2.34x speedup for math-calc task** and **2.29x speedup for longbook-sum-eng task** with Llama-3.1-8B model (prefill length=32k, batch size=128, draft budget=2k, gamma=4) on 8 A100s. Both these speedups are higher than what we achieved for PG-19 task on 8 H100s.
>
> 2. We also implemented SnapKV-based self-speculation on the state-of-the-art LLM inference framework MLC-LLM, and compared the end-to-end speedup results in $\text{\textcolor{blue}{ (Table 4 and Table5, Section A.3, Page 14)}}$, which also highlights the generalizability of our method.

---

> ### Author Response · Authors · 2024-12-03
>
> Dear Reviewer XfUb,
>
> If our latest responses have addressed your additional concerns, we kindly ask you to consider adjusting your rating back to positive. Thanks for your time and valuable insights.

---

### Author Response · Authors · 2024-11-21
**Manuscript Revision Summary**

We thank reviewers **[R1 (XfUb), R2 (PRVw), R3 (UdZh), R4 (3wSP)]** for their thoughtful and highly supportive feedback! We were glad that the reviewers found the problem **significant and interesting [R1, R2, R3, R4]**, the observations and theoretical analysis **insightful and highly valuable [R1, R2, R3, R4]**, the methods **novel and clever [R3]**, the presentation **easy to follow [R1, R3, R4]**, and the experimental results **strong and impressive [R1, R3]**.

We have updated the paper to incorporate constructive suggestions, which will be shown in the revision. Below is a summary of the major changes:

---

### **[R1, R2] Contribution of MagicDec**
- We have updated the Introduction section to clarify the main contribution of our work. MagicDec is the **first approach** to demonstrate that speculative decoding can improve speedup even for large batches when serving long-context sequences.
- Through analysis of speculative decoding speedup and LLM inference performance, we identify that the key to achieving this is ensuring the draft cost grows independently of sequence length.
- KV cache compression is the key method that limits draft cost while maintaining high acceptance rates.
- MagicDec provides a general framework to guide the choice of optimal drafting strategies or KV compression methods based on draft cost, acceptance rate, and hardware.
  $\text{\textcolor{blue}{(Section 1, Pages 1-2)}}$

---

### **[R2] Comparison with Normal Speculative Decoding**
- We have added experiments demonstrating that standard speculative decoding does not perform well in short context-length, large batch size regimes $\text{\textcolor{blue}{(Fig. 7a, Page 10)}}$.
- For moderate context lengths, standard speculative decoding with a small draft model using a full KV cache achieves some speedup but fails to scale with batch size.
- Conversely, applying KV compression to the small draft model significantly improves speedup by reducing draft cost $\text{\textcolor{blue}{(Fig. 7b, Page 10)}}$.
- For long context, large batch size regimes, self-speculation with KV compression emerges as the optimal drafting strategy due to its high acceptance rate $\text{\textcolor{blue}{(Fig. 7c, Page 10)}}$.
$\text{\textcolor{blue}{(Section 5.3, Page 10)}}$

---

### **[R1] Generalization to Different Model Families**
- We have added experimental results and discussions for the Qwen2.5-7B, Qwen2.5-32B, and Mistral-7B-v0.3 models in the Evaluation section.   $\text{\textcolor{blue}{(Appendix A.5, Page 15)}}$
- These results demonstrate that speculative decoding with compression performs well across these models, showing trends similar to those observed in Llama models.
- This further validates the effectiveness of our analysis.
  $\text{\textcolor{blue}{(Section 5.3, Page 10)}}$

---

### **[R3] More Streamlined Explanation of Theoretical Analysis**
- We revised the sections discussing KV cache bottlenecks and speculative decoding speedup factors to improve readability.
  $\text{\textcolor{blue}{(Sections 3.2, 3.3, Pages 4-6)}}$

---

### **[R2] Explanation of How MagicDec Selects the Optimal Drafting Strategy**
- Section 4 has been restructured to more effectively explain the key aspects of KV compression-based drafting when selecting the ideal drafting method.  We independently analyze the three main factors that affect MagicDec's performance - draft model size, draft model KV cache size and KV compression algorithm.
  $\text{\textcolor{blue}{(Section 4, Pages 6-8)}}$

---

### **[R1] Limitations Discussion**
- We have added a discussion of the limitations and future work for MagicDec in the Conclusion section.
  $\text{\textcolor{blue}{(Section 6, Page 10)}}$

---

### **[R2, R3] Latency Breakdown**
- Fig. 1(a) has been updated to isolate KV load time and KV store time for greater clarity.
- The inference bottleneck under large batch size and long sequence regimes is specifically **KV load time**, with KV store time constituting only a small portion of total inference time.
- This breakdown is inferred from **[1]**.
  $\text{\textcolor{blue}{(Section 1, Pages 1-2)}}$

---

### **References**
- **[1]** Yuan Z., Shang Y., Zhou Y., et al. *LLM Inference Unveiled: Survey and Roofline Model Insights*. arXiv preprint arXiv:2402.16363, 2024.

---

### Meta-Review · Area_Chair_G5Qy · 2024-12-21

**Metareview:**

(a) Summary of Scientific Claims and Findings

The paper presents MagicDec, a speculative decoding technique aimed at improving throughput and reducing latency for long-context Large Language Models (LLMs). It challenges the conventional understanding by demonstrating that speculative decoding can be effective even in high-throughput scenarios with large batch sizes and extended sequences, achieved through an optimized drafting strategy and sparse Key-Value (KV) caching.

(b) Strengths of the Paper

1. The paper challenges the traditional view that speculative decoding is only viable for small batch sizes.

2. MagicDec demonstrates compatibility with a wide range of LLMs and hardware setups.

3. It achieves notable speedup without compromising accuracy.

4. The framework is versatile, supporting multiple drafting strategies and KV compression techniques.

(c) Weaknesses of the Paper and Missing Elements

1. Initial reviews noted a lack of comparisons with related works and speculative decoding baselines (e.g., TriForce, SWIFT), along with limited evaluations on diverse LLM families and datasets (e.g., InfiniteBench). Most of these issues were addressed during the discussion phase.

2. Some sections, such as the theoretical explanation of KV caching, are overly dense and could benefit from improved clarity.

3. The paper focuses exclusively on decoding performance, leaving prefill optimization unexplored.

(d) Decision and Rationale

The paper’s strengths outweigh its weaknesses. It offers novel insights that question established beliefs about speculative decoding, provides solid experimental evidence with practical implications for LLM serving systems, and effectively addresses most reviewer concerns through constructive revisions.

**Additional Comments On Reviewer Discussion:**

The authors resolved most issues by presenting additional experimental results and clarifying critical aspects of their methodology. They also emphasized how MagicDec complements existing approaches, strengthening its relevance and applicability.

---

### Decision · Program_Chairs · 2025-01-22

Accept (Poster)